# One-shot analysis of translated mammalian lncRNAs with AHARIBO

**Luca Minati[1†], Claudia Firrito[1†], Alessia Del Piano[1†], Alberto Peretti[1†], Simone Sidoli[2], Daniele Peroni[3], Romina Belli[3], Francesco Gandolfi[4], Alessandro Romanel[4], Paola Bernabo[1], Jacopo Zasso[5], Alessandro Quattrone[5], Graziano Guella[6], Fabio Lauria[7], Gabriella Viero[7], Massimiliano Clamer[1]\***

[1]IMMAGINA BioTechnology, Trento, Italy; [2]Department of Biochemistry, Albert Einstein College of Medicine, Bronx, United States; [3]Mass Spectrometry Facility, Computational and Integrative Biology (CIBIO), University of Trento, Trento, Italy; [4]Laboratory of Bioinformatics and Computational Genomics, Department of Cellular, Computational and Integrative Biology (CIBIO), University of Trento, Trento, Italy; [5]Laboratory of Translational Genomics, Department of Cellular, Computational and Integrative Biology (CIBIO), University of Trento, Trento, Italy; [6]Department of Physics, University of Trento, Trento, Italy; [7]Institute of Biophysics, CNR Unit at Trento, Trento, Italy

**Abstract** A vast portion of the mammalian genome is transcribed as long non-coding RNAs (lncRNAs) acting in the cytoplasm with largely unknown functions. Surprisingly, lncRNAs have been shown to interact with ribosomes, encode peptides, or act as ribosome sponges. These functions still remain mostly undetected and understudied owing to the lack of efficient tools for genome-wide simultaneous identification of ribosome-associated and peptide-producing lncRNAs. Here, we present AHA-mediated RIBOsome isolation (AHARIBO), a method for the detection of lncRNAs either untranslated, but associated with ribosomes, or encoding small peptides. Using AHARIBO in mouse embryonic stem cells during neuronal differentiation, we isolated ribosome-protected RNA fragments, translated RNAs, and corresponding de novo synthesized peptides. Besides identifying mRNAs under active translation and associated ribosomes, we found and distinguished lncRNAs acting as ribosome sponges or encoding micropeptides, laying the ground for a better functional understanding of hundreds of lncRNAs.

*For correspondence:
mclamer@immaginabiotech.com

†These authors contributed equally to this work

## Introduction

An incredibly small fraction of the mammalian genome is protein-coding (<3%), while the number of potentially functional non-coding genes remains unclear (*Djebali et al., 2012*). Long non-coding RNAs (lncRNAs) are defined as non-coding RNA exceeding 200 nt. They have gained much attention because of their role in a variety of cellular processes, from chromatin architecture (*Minajigi et al., 2015*) to mRNA turnover (*Kleaveland et al., 2018*) and translation (*Ingolia et al., 2011*). Typically, lncRNAs are abundant transcripts (*Iyer et al., 2015*) that display short and not evolutionarily conserved Open Reading Frames (ORFs with minimal homology to known protein domains (*Guttman and Rinn, 2012*). The majority of lncRNAs are localized in the cytoplasm (*Carlevaro-Fita et al., 2016*), where they are supposed to remain untranslated. Ribosome profiling (RIBO-seq), which provides positional information of ribosomes along transcripts (*Clamer et al., 2018*; *Ingolia et al., 2012*), identified several ribosome-associated lncRNAs (*Bazzini et al., 2014*; *Ingolia et al., 2011*; *Lee et al., 2012*; *Zeng et al., 2018*). A handful of lncRNAs have been shown to be involved in translation regulation (*Carrieri et al., 2012*; *Yoon et al., 2012*), while others are

themselves potentially or partially translated (*Anderson et al., 2015*; *Aspden et al., 2014*; *Bazin et al., 2017*; *Ingolia et al., 2011*; *Nelson et al., 2016*; *Ruiz-Orera et al., 2014*; *van Heesch et al., 2019*). As coding RNAs, lncRNAs can be associated with actively translating or translationally silent ribosomes (*Chandrasekaran et al., 2019*; *Chen et al., 2020*; *Jiao and Meyerowitz, 2010*; *Kapur et al., 2017*). Hence, the potential involvement of lncRNAs in translation increases the complexity of the mammalian control of gene expression at the translatome and proteome level. Unfortunately, classical RIBO-seq approaches barely distinguish between lncRNAs producing peptides from those that sequester ribosomes (lncRNA bound to ribosomes without translation) and act as ribosome sponges. Proteomics approaches, such as mass spectrometry, can help to define and quantitatively monitor the production of peptides, but are less sensitive techniques than RNA sequencing (*Slavoff et al., 2013*; *van Heesch et al., 2019*). Therefore, proteomics and RIBO-seq alone cannot unravel the wide functional range of cytoplasmic lncRNAs associated with the translation machinery.

To fill this gap, we developed AHA-mediated RIBOsome isolation (AHARIBO), a combination of protocols that simultaneously isolate RNAs and nascent proteins associated with translationally active ribosomes. AHARIBO is based on the isolation of ribosomes trapped with their nascent peptides by incorporating the non-canonical amino acid L-azidohomoalanine (AHA), followed by parallel RNA-seq, ribosome profiling, and proteomics.

We applied AHARIBO to human and mouse cells and showed that it enables to (1) purify translating ribosomes via nascent peptide chains, (2) co-purify RNAs and proteins for transcriptome/de novo proteome-associated studies, and (3) detect the regulatory network of lncRNAs translated or associated with ribosomes.

## Results

### Nascent peptide labeling and separation of the ribosome complex with AHARIBO-rC

To simultaneously purify ribosomes under active translation, associated RNAs, and corresponding growing peptide chains, we optimized a protocol in HeLa cells (*Figure 1A*). Briefly, the protocol consists of the following phases: (1) incubation with a methionine-depleted medium, (2) addition of the methionine analog AHA, (3) on-ribosome anchorage of nascent peptide chains with a small molecule, (4) cell lysis and AHA 'copper-free click reaction' (*Jewett and Bertozzi, 2010*) for (5) ribosome capture with magnetic beads. We reasoned that the protocol for isolating ribosomes through AHA can be used to obtain information about nascent peptides, constitutive components of ribosomes, mRNAs, and lncRNAs associated with them. For this reason, we optimized several parameters from washing steps to nuclease treatments (*Figure 1A*) to isolate (1) the full translational complex (AHARIBO-rC, *r*ibosomal *c*omplexes: ribosomes, ribosome-associated proteins, nascent peptides, and RNAs), (2) the de novo synthesized proteome (AHARIBO-nP, *n*ascent *p*roteome), and (3) ribosome-protected fragments (RPFs) (AHARIBO RIBO-seq: *RIBO*some profiling by *seq*uencing).

To minimize the amount of AHA-tagged and fully synthesized proteins released from ribosomes and achieve optimal on-ribosome polypeptide stabilization, we tested multiple incubation times of AHA exposure and compared the effect of two small molecules (namely cycloheximide [CHX] and sBlock, an anisomycin-based reagent). Anisomycin is known to inhibit the activity of eukaryotic ribosomes, while keeping polypeptides bound to translating ribosomes (*Garreau de Loubresse et al., 2014*; *Grollman, 1967*; *Seedhom et al., 2016*).

We observed that 30 min is the optimal incubation time for sufficient AHA incorporation and maximum RNA recovery (*Figure 1—figure supplement 1A–C*). Next, we compared the efficiency of CHX and sBlock in stabilizing the nascent peptide by co-sedimentation analysis of AHA-tagged polypeptides with ribosomes along the sucrose gradient (*Figure 1B*). As a control, cells were treated in parallel with puromycin to cause ribosome disassembly and release of the growing peptide chains (*Figure 1B*; *Blobel and Sabatini, 1971*; *Enam et al., 2020*). In agreement with literature, we found that both CHX and sBlock are able to stabilize AHA-peptides on ribosomes and polysomes (*Biever et al., 2020*; *Mathias et al., 1964*). The efficiency of anchoring polypeptides on ribosomes in CHX- and sBlock-treated cells was about 50% higher compared to untreated cells, confirming that the treatment effectively stabilizes nascent polypeptides (*Figure 1C*). The high signal observed in

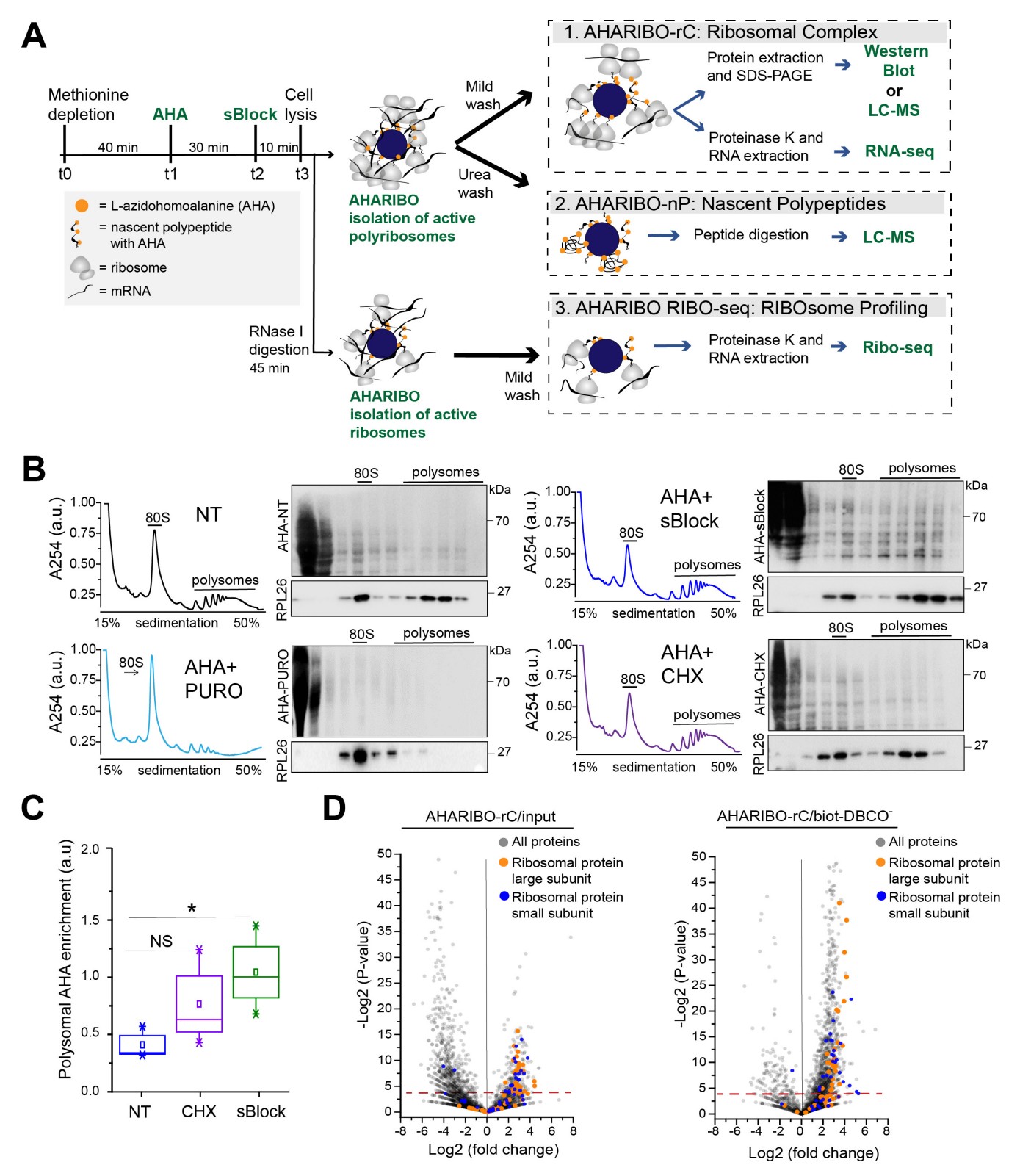

**Figure 1.** L-Azidohomoalanine (AHA) labeling of nascent peptide chains and ribosome separation. (**A**) Schematic representation of AHA-mediated RIBOsome isolation (AHARIBO) workflow. After methionine depletion, AHA incubation, and sBlock treatment, cell lysates can be processed for (1) AHARIBO-rC: isolation of translational complexes (ribosomes, ribosome-associated proteins, nascent peptides, and RNAs); (2) AHARIBO-nP: isolation of de novo synthesized proteome; and (3) AHARIBO RIBO-seq: for ribosome profiling. (**B**) Polysomal profiles in HeLa cells. On the right of each

*Figure 1 continued on next page*

*Figure 1 continued*

profile, example of SDS-PAGE of protein extracts from each fraction of the profile. Staining of the membrane was performed by biotin cycloaddition followed by streptavidin-Horseradish peroxidase (HRP). RPL26 protein was used as a marker of the large ribosome subunit. (C) Box plot showing the AHA signal enrichment in the polysomal fractions of the profiles in cells untreated (NT) and treated with either cycloheximide (CHX) or sBlock. Results are shown as the median (±SE) of three independent experiments. NS: not significant. *p-value=0.05 was obtained through an unpaired t-test. (D) Volcano plots of AHARIBO-rC-isolated proteins. Data are compared with input (AHA-containing lysate, left) or with streptavidin-coated beads without biotin-DBCO (right). DBCO: dibenzocyclooctyne. Red line: t-test p-value<0.05.

The online version of this article includes the following source data and figure supplement(s) for figure 1:

**Source data 1.** A table with the relative abundance of AHARIBO-rC-isolated proteins.
**Source data 2.** Gene Ontology analysis data.
**Figure supplement 1.** L-Azidohomoalanine (AHA) incorporation, validation of AHA, and RNA capture.
**Figure supplement 2.** Liquid chromatography-mass spectrometry (LC-MS) analysis of AHARIBO-rC proteins and validation by western blot.
**Figure supplement 3.** AHARIBO-rC efficiency test and validations.

lighter fractions is likely caused by AHA-labeled proteins released from ribosomes. To overcome this problem, it is possible to perform a pre-cleaning of the cell lysate by sucrose cushioning. This step can increase the efficiency of total RNA isolation with AHARIBO compared with the control (no AHA) (*Figure 1—figure supplement 1D*). As expected, in puromycin-treated samples, the AHA signal was mainly detected in the first two fractions of the gradient, proving that the signal observed in the heavier fractions of CHX- and sBlock-treated cells was not caused by diffusion of AHA-labeled peptides from lighter to heavier fractions. Since sBlock outperformed CHX in anchoring efficiency (*Figure 1C*), we used this compound in all further experiments.

Prompted by the evidence that nascent peptides can be stably anchored on ribosomes by a small molecule, we isolated RNAs and proteins associated with the translation complex. To this aim, we performed a label-free liquid chromatography-mass spectrometry (LC-MS) analysis of AHARIBO-captured proteins relative to the input, to the background biotin-DBCO$^-$ (*Figure 1D*) or AHA$^-$ (*Figure 1—figure supplement 2A*; *Figure 1—source data 1*) and to a sample treated with puromycin (AHA$^+$ puromycin) (*Figure 1—figure supplement 2B*), which causes the release of nascent chains. We observed that ribosomal proteins belonging to both the large and small ribosome subunits are indeed more abundant in AHARIBO-rC samples than in controls. LC-MS results were confirmed by western blot analysis of proteins that are component of the large and small ribosomal subunits (RPS6, RPL26) (*Figure 1—figure supplement 2B*). Gene ontology (GO) analysis revealed that terms related to translation (biological process), nucleic acid binding (cellular function), and ribonucleoprotein complex (cellular component) are enriched in AHARIBO-rC compared to the control (no AHA), confirming efficient pulldown of translation-related proteins (*Figure 1—source data 2*).

Then, we used AHARIBO-rC to determine the translational status of cultured cells. To this aim, we downregulated protein synthesis by treating HeLa cells with puromycin, heat shock (HS) (10 min at 42°C, during AHA incubation), or arsenite (Ar) treatment, which induces translational inhibition and stress granules formation (*Wang et al., 2016*). We observed a reduction of RNA captured in puromycin-, HS-, and Ar-treated cells relative to the control (*Figure 1—figure supplement 3A–C*). In line with this finding, qRT-PCR analysis showed about 50% reduction in 18S rRNA levels when translation was inhibited (*Figure 1—figure supplement 3D*).

To further validate AHARIBO-rC, we took advantage of a micropeptide (176 aa) originating from an open reading frame of the TUG1 lncRNA, called TUG1-BOAT (*Lewandowski et al., 2020*). The wild-type (WT) ORF has a non-canonical start codon and a methionine 75 nt upstream of the stop codon. We ectopically expressed the WT TUG1-BOAT transcript and two mutant constructs (*Figure 1—figure supplement 3E*): (1) the ΔTUG1-BOAT, without the methionine 75 nt upstream of the stop codon and (2) the +1Met TUG1-BOAT with an ATG (methionine) as start codon. The +1Met TUG1-BOAT has two methionines, one at the N terminal and the other at 25 aa (75 nt) before the C-terminal. Our RT-qPCR analysis performed 24 hr or 48 hr after transfection showed a good efficiency of AHARIBO in capturing the TUG1-BOAT RNA when methionines are present (about 50 times more in +1Met TUG1-BOAT than in ΔMet TUG1-BOAT after 24 hr) (*Figure 1—figure supplement 3E*), confirming the efficiency of AHARIBO-rC in capturing translated RNA.

## AHARIBO-nP: genome-wide portray of the de novo synthesized proteome

Motivated by the evidence that AHARIBO-rC can be used to isolate bona fide active ribosomes, we further tested our method genome-wide in mouse embryonic stem cells (mESCs) under basal condition and after differentiation into early neurons (ENs) (*Tebaldi et al., 2018*; *Figure 2—figure supplement 1A*). We analyzed both AHARIBO-rC-isolated RNA and newly synthesized polypeptides associated with actively translating ribosomes by RNA-seq and LC-MS, respectively. The protocol for the isolation of the de novo synthesized polypeptides (named AHARIBO-nP) is based on urea washing to remove all proteins that are not nascent peptides (*Figure 2—figure supplement 1B*). In parallel, we isolated and analyzed the global translatome by extracting the RNA after 30% sucrose cushioning of cytoplasmatic lysates (*Wang et al., 2013*), and then analyzed the global proteome by pulsed SILAC (pSILAC) (*Schwanhäusser et al., 2009*; *Figure 2A*).

Quantitative proteomic analysis of ENs versus mESCs (EN/mESC) led to the identification of 2654 differentially expressed proteins (*Figure 2B*, *Figure 2—source data 1*). As expected, differentiated cells (EN) showed a reduced turnover compared to mESCs (*Figure 2—figure supplement 1C*). In parallel, EN and mESC cells were analyzed by AHARIBO-nP, which captured 1365 and 2215 proteins, respectively. Of note, 74% of proteins identified through AHARIBO-nP is in common with the pSILAC dataset. The smaller number of proteins identified with AHARIBO-nP compared to pSILAC is most probably related to the shorter time of incubation with AHA (30 min) compared to pSILAC (24 hr) and is consistent with previous observations from similar pulldown enrichment strategies (*Bagert et al., 2014*; *Rothenberg et al., 2018*). Differential expression analysis (EN/mESC) identified 573 proteins (p-value<0.05) in AHARIBO-nP (*Figure 2B*; *Figure 2—source data 2*). The GO analysis of differentially expressed proteins showed that proteins involved in cytoskeleton organization and neurogenesis were upregulated (*Figure 2C*), further confirming the reliability of AHARIBO-nP in monitoring de novo protein expression. We focused on proteins captured by AHARIBO-nP during differentiation (*Figure 2C*, *Figure 2—source data 2*) and found that several are known to be expressed during early stages of development of the nervous system (e.g., Map1b, Tubb3, and Dync1h1) (*Fiorillo et al., 2014*; *Gonzalez-Billault et al., 2002*; *Latremoliere et al., 2018*). In addition, we performed AHARIBO-nP pulldown in mESCs double-labeled for pSILAC (24 hr) and AHA (30 min) (*Figure 2D*). Interestingly, we observed high fold changes of heavy amino acids in AHARIBO-nP (*Figure 2E*) and a significantly higher protein turnover in the AHARIBO-nP compared to the pSILAC proteins (*Figure 2—figure supplement 1D*), suggesting that AHARIBO-nP is indeed able to capture the de novo synthesized polypeptides.

Collectively, these results show that AHARIBO-nP captures de novo synthesized proteins and produces meaningful descriptions of phenotypic changes occurring upon cell differentiation. Moreover, these results demonstrate that our AHARIBO-nP protocol is suitable to monitor dynamic changes in protein expression by LC-MS analysis.

## Combination of AHARIBO-rC and AHARIBO-nP: parallel genome-wide analysis of translated RNAs and de novo synthesized proteome

Prompted by previous results, we asked if mRNAs purified using AHARIBO-rC are a good proxy of protein levels. To this aim, we compared AHARIBO-rC RNA and the global translatome with AHARIBO-nP in mESCs during differentiation.

To exclude any bias related to protein length, we checked whether AHARIBO-nP preferentially captures long or short proteins. We plotted the peptide size against the enrichment resulting from AHARIBO-rC compared with the global transcriptome (*Figure 3A*). This value represents the extent to which AHARIBO-rC RNA differs from the standard method. Our results confirm that AHARIBO captures transcripts encoding for polypeptides in a wide range of length (*Figure 3A*). Since in all eukaryotes proteins are initiated with a methionine residue and the average protein size in eukaryotes is about 300 aa (*Frith et al., 2006*), virtually any protein can be captured as soon as the nascent peptide exits the ribosome (i.e., when it reaches a length of about 35–40 aa). In about 70% of the proteome, the N-terminal methionine is co-translationally cleaved when the peptide is at least 50 aa long by the enzyme methionine aminopeptidase (*Wild et al., 2020*), while the remaining 30% retains the methionine (*Martinez et al., 2008*). Therefore, there is a reasonable probability for at least one AHA residue to be available for each peptide when the inhibitor of translation (sBlock) is

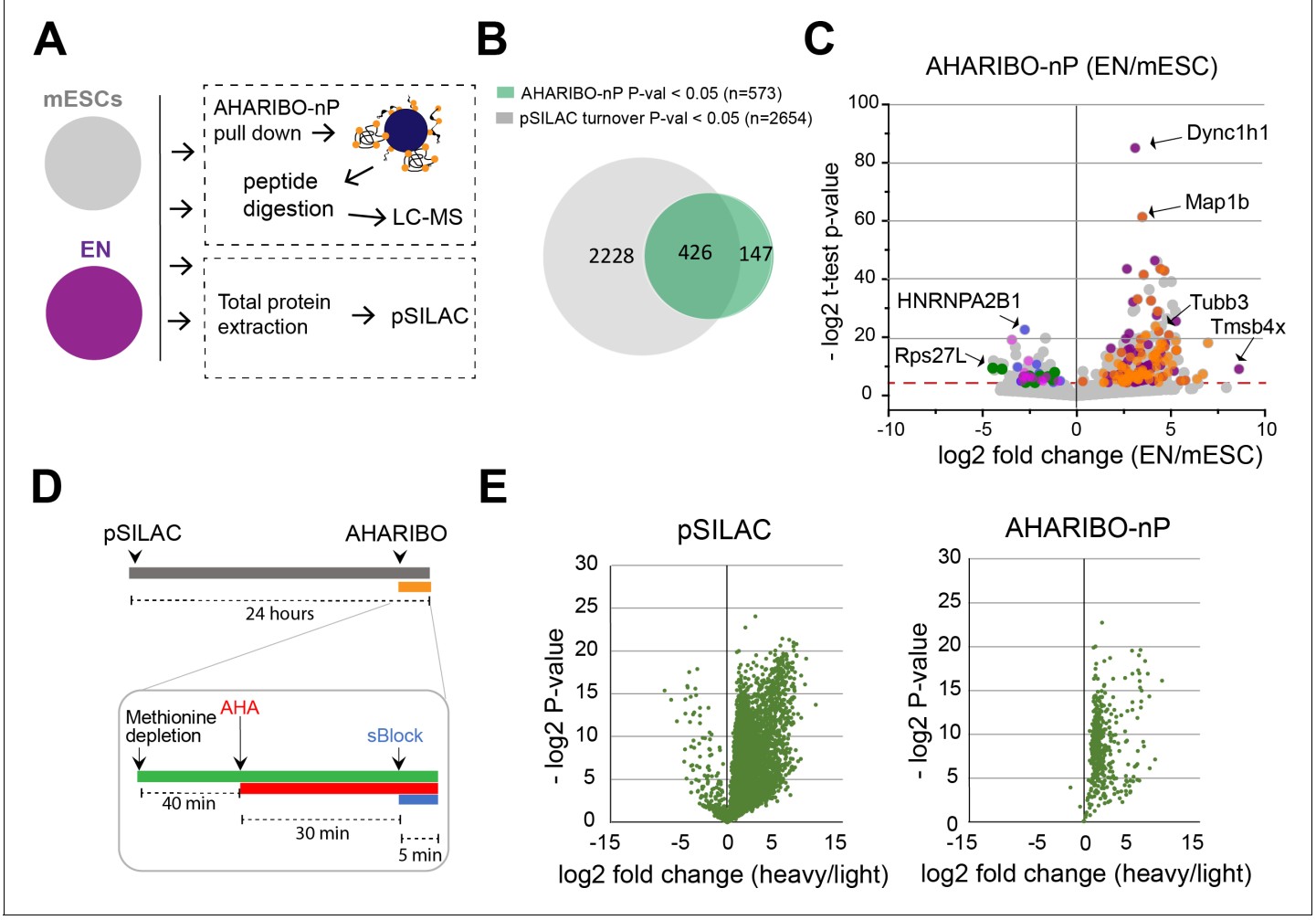

**Figure 2.** AHARIBO-nP and pSILAC. (**A**) Workflow for parallel AHARIBo-nP and pSILAC. mESCs: mouse embryonic stem cells; EN: mouse embryonic stem cells differentiated in early neurons. (**B**) Venn diagram representing the number of differentially expressed proteins (EN/mESCs) identified by AHARIBO-nP and pSILAC (p-value<0.05). (**C**) Volcano plot for each differentially expressed protein (EN/mESC) of AHARIBO-nP proteome versus -log2 (p-value). Red broken line indicates p-value<0.05. Orange and purple dots represent upregulated proteins involved in cytoskeleton organization (GO:0007010) and neurogenesis (GO:0022008), respectively. Blue, green, and magenta dots represent downregulated proteins related to RNA processing (GO:0006396), protein synthesis (GO:0006412), and mouse pluripotency (WP1763). Gray dots represent all other proteins. (**D**) Schematic representation of combined cell treatments for pSILAC and AHARIBO-nP. (**E**) Volcano plots displaying for each protein the -log2 t-test p-value against the fold changes of protein turnover (heavy/light) in pSILAC proteome (left) and AHARIBO-nP (right) for double-treated mESCs. GO: gene ontology; AHARIBO: AHA-mediated RIBOsome isolation; pSILAC: pulsed SILAC.

The online version of this article includes the following source data and figure supplement(s) for figure 2:

**Source data 1.** A table with the pulsed SILAC (pSILAC) proteomic data.
**Source data 2.** A table with AHA-mediated RIBOsome isolation (AHARIBO) differentially expressed proteins.
**Figure supplement 1.** Cell differentiation and additional proteomic analysis.

added to the cell medium, enabling the capture of the polypeptide outside the ribosome exit tunnel.

To further prove the reliability of our method, we measured the efficiency of AHARIBO-rC to capture coding transcripts compared to a global translatome analysis. Using increasing abundance thresholds in EN, we observed that AHARIBO-rC efficiency is comparable to the global translatome for low abundant transcripts in EN and for all transcripts in undifferentiated mESCs (*Figure 3—figure supplement 1A*). Strikingly, AHARIBO captures abundant transcripts in EN with much higher efficiency than the global translatome (*Figure 3B*).

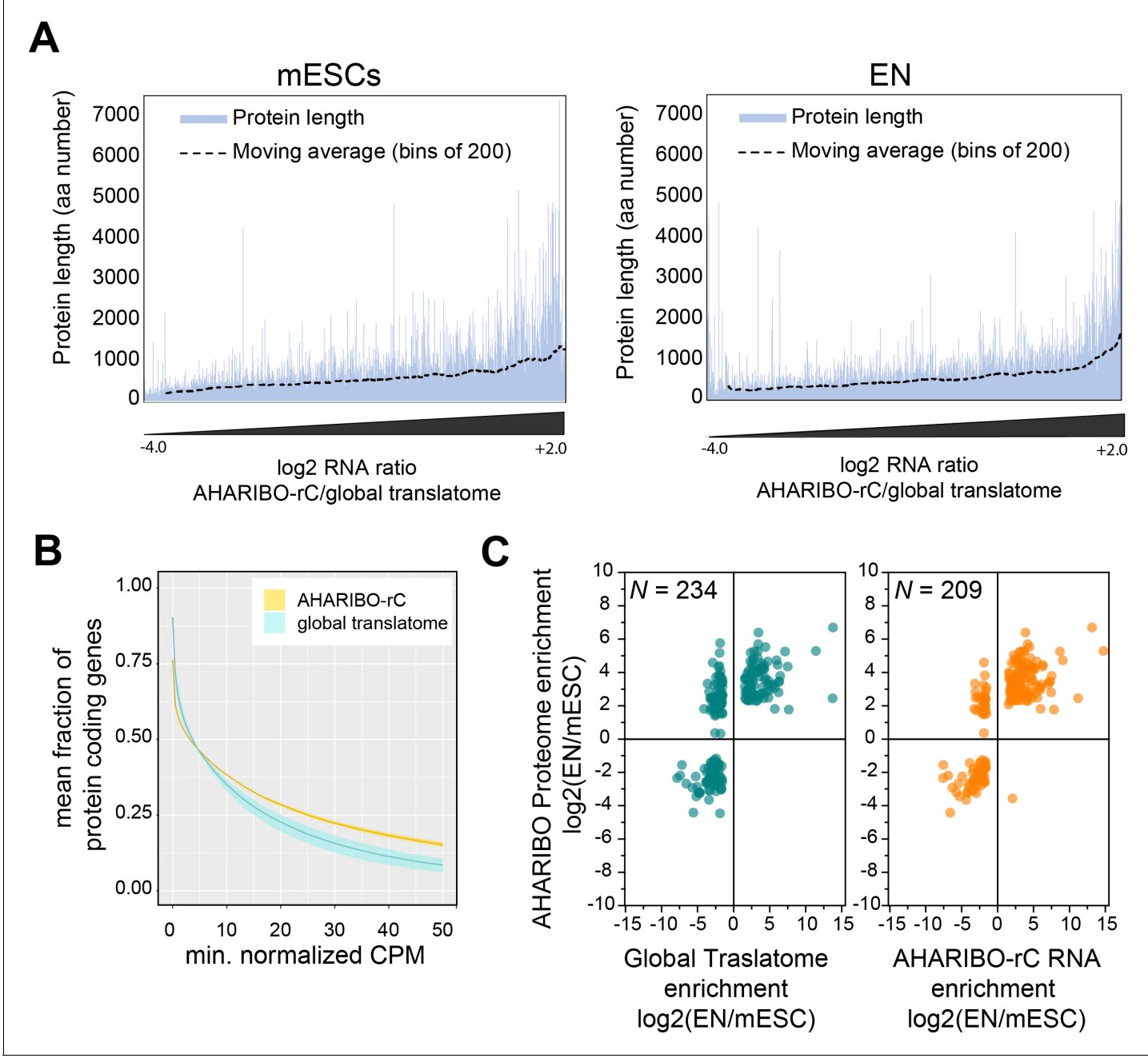

**Figure 3.** AHARIBO-rC RNA versus de novo proteome analysis. (**A**) Enrichment of a given transcript obtained with AHA-mediated RIBOsome isolation (AHARIBO) versus global translatome (x-axis) as a function of the theoretical protein length (y-axis) for mouse embryonic stem cells (mESCs) (left) and early neurons (ENs) (right). Each bar represents the number of enriched transcripts with the defined theoretical protein length. (**B**) Fraction of coding genes expressed above a minimum threshold in EN. The AHARIBO-rC and global translatome group are represented in yellow and cyan, respectively. For each group, the mean (solid line) and SD (shades) of the fractions for a given count per million (CPM) threshold are calculated over all samples (n = 6) in that group. (**C**) Scatter plot of RNA fold change (global translatome on the left, AHARIBO-rC on the right) compared to protein fold change (AHARIBO-nP) obtained by comparing EN with mESC. N: number of differentially expressed genes (DEGs) with p-value<0.05.
The online version of this article includes the following source data and figure supplement(s) for figure 3:

**Source data 1.** A table with differentially expressed genes (DEGs) from RNA-seq data comprising logFC, LogCPM, LogFWER, and LogPval.
**Source data 2.** A table with RNA and protein differentially expressed genes (DEGs) from AHARIBO-nP, pSILAC, AHARIBO-rC, and global translatome.
**Figure supplement 1.** RNA-seq and protein coding RNA analysis.

Finally, we tested whether the RNA isolated with AHARIBO-rC can predict the de novo synthesized proteome. After comparing differentially expressed genes (DEGs) during differentiation to the AHARIBO-nP proteome (*Figure 3—source data 1*), we observed that AHARIBO-rC RNA is a good proxy of the newly synthesized proteome (Pearson's correlation *r* = 0.75, *Figure 3C*, *Figure 3—figure supplement 1B*). In particular, we found that AHARIBO-rC RNA presents less uncoupled genes (up-RNA and down-protein or down-RNA and up-protein) than the global translatome (*Figure 3—figure supplement 1C*), thus faithfully recapitulating proteome changes. The correlation of the global translatome with the global protein turnover measured with pSILAC shows a Pearson's *r* = 0.27 (*Figure 3—figure supplement 1D*, *Figure 3—source data 2*). This result demonstrates that AHARIBO-nP does reflect the labeling of peptides rather than completely synthesized proteins.

## Combined AHARIBO approaches define the functional role of lncRNAs in translation

Based on the evidence that a combination of AHARIBO approaches can simultaneously detect RNAs under active translation and peptides in the process of being produced, we applied our methods to detect ribosome-associated and translated native lncRNAs.

In AHARIBO-rC data, we identified a total of 687 lncRNA genes in mESCs and about 400 differentially expressed (DE) lncRNAs during neuronal differentiation (*Figure 4—figure supplement 1A*, *Figure 4—source data 1*). Among the top five DE lncRNAs (fold change >10; p-value<$1\times10^{-10}$), we found *Pantr1* and *Lhx1os*, known to be involved in neuronal development (*Biscarini et al., 2018*; *Carelli et al., 2019*). To identify potentially translated lncRNAs, we applied the abundance threshold analysis to the subset of AHARIBO-rC non-coding RNAs in common with a published dataset (n = 270) of lncRNA identified by ribosome profiling data in mESCs (*Ingolia et al., 2011*; *Figure 4—figure supplement 1B*). The analysis of 100 lncRNAs in common between the two datasets showed a stronger enrichment of ribosome footprints in the AHARIBO-rC than in the global translatome (*Figure 4A, Figure 4—figure supplement 1C*). Altogether, these results suggest that a fraction of non-coding transcripts, which is efficiently isolated with AHARIBO-rC, is potentially translated.

To understand if and how lncRNAs interact with ribosomes, we performed ribosome profiling experiments after AHARIBO pulldown (named AHARIBO RIBO-seq), with parallel standard RNA-seq (on inputs) analysis in mESCs. For protein-coding genes, both standard and AHARIBO RIBO-seq show an enrichment of RPFs in the coding sequence (*Figure 4—figure supplement 2A*). The two datasets show high correlation (*Figure 4—figure supplement 2B*) and the expected codon periodicity in the coding sequence in AHARIBO RIBO-seq (*Figure 4—figure supplement 2C*). These results further confirm the capability of AHARIBO in capturing ribosomes. With AHARIBO RIBO-seq, we identified a list of lncRNAs covered by ribosome footprints (*Figure 4—source data 2*). By intersecting our AHARIBO RIBO-seq data with those obtained from standard methods (RIBO-seq and RNA-seq after sucrose cushioning) or AHARIBO-rC, we identified 125 common putative translated lncRNAs (*Figure 4B*). Some of these lncRNA (n = 19) are known to be translated in mouse tissue (*van Heesch et al., 2019*). The vast majority of these lncRNAs do not have a known function. Two of the identified lncRNAs (9330151L19Rik and Gm9776) were detected only by standard RIBO-seq and RNA-seq but not with AHARIBO (*Figure 4C*). This result may be due to the absence of translation events (i.e., transcripts loaded with idle ribosomes). Next we validated the coding potential of lncRNAs that are in common between AHARIBO and standard RIBO-seq (*Figure 4D*). We translated in silico the transcripts in all frames to find potential ORFs with a canonical start codon (AUG). Translated sequences were semi-trypsin-digested in silico and then manually annotated to find confident matching spectra from the AHARIBO-nP protein dataset. Out of the about 46,000 collected spectra (*Figure 4—source data 3*), our MS-based proteomics analysis detected peptides with highly corresponding ribosome footprints (e.g., Gm42743, Gm26518, B230354K17Rik, D030068K23Rik, 1810058I24Rik). From the list of 129 lncRNAs that are in common among all AHARIBO protocols and standard RIBO-seq (*Figure 4D*), we identified by MS analysis a micropeptide (Mm47) of 47 aa (*Figure 4D*) at a high degree of confidence. This micropeptide derives from a lncRNA expressed in murine macrophages, and recently characterized by an independent group (*Bhatta et al., 2020*) as a relevant peptide able to modulate the innate immunity in mice. Several other lncRNAs show high confidence of translation events with in silico prediction even if they were not perfectly matching our proteomic spectra (*Figure 4—figure supplement 3*), paving the way for a better characterization of translatable lncRNA that has not been reported before. These results, combined with (1) AHARIBO's

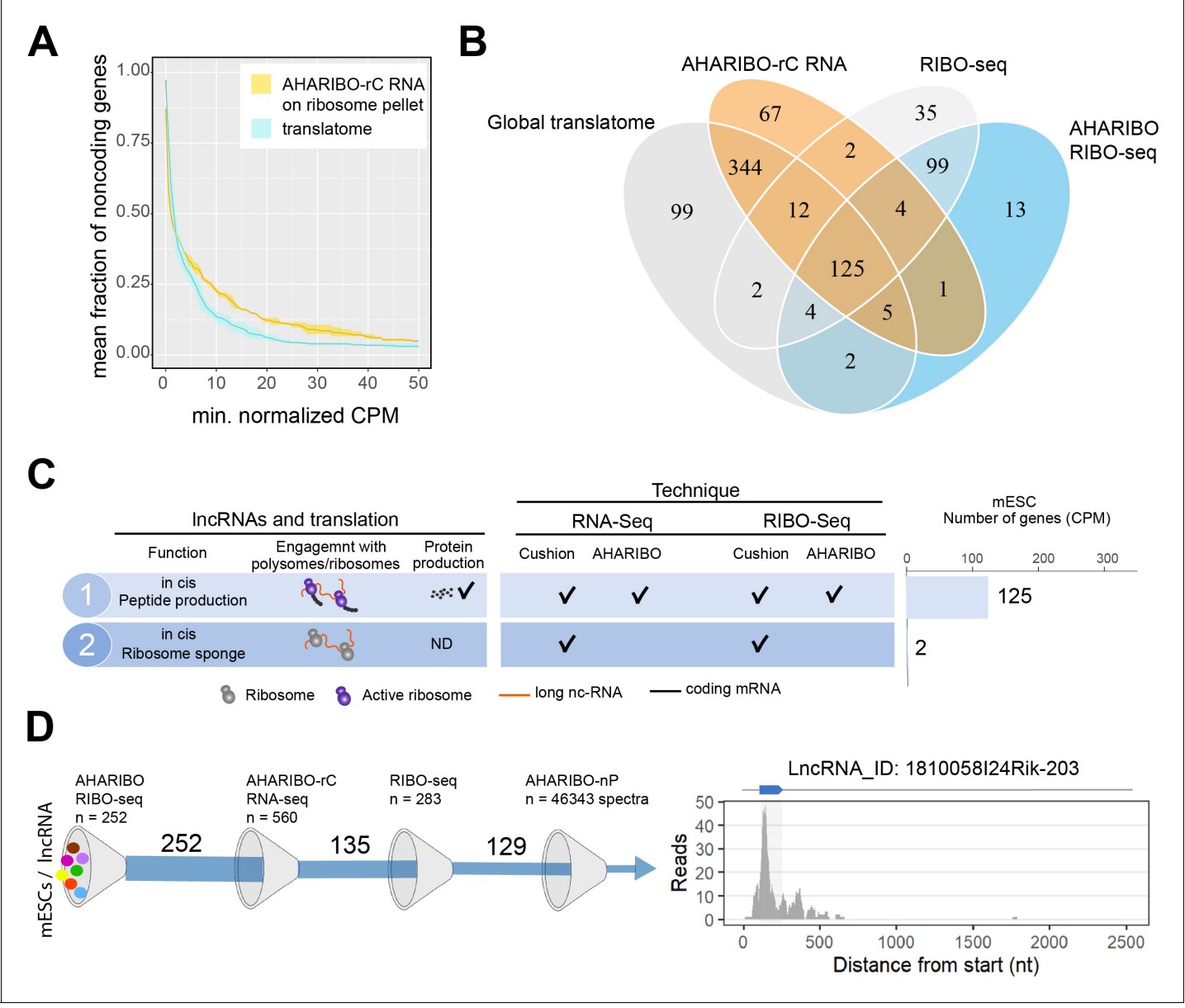

**Figure 4.** The AHA-mediated RIBOsome isolation (AHARIBO) platform can be used to detect ribosome-interacting long non-coding RNAs (lncRNAs). (**A**) Linear plot illustrating the fraction of non-coding genes expressed above a minimum threshold in early neurons (EN). The AHARIBO-rC and the global translatome group are represented in yellow and cyan, respectively. For each group, the mean (solid line) and the SD (shades) of the fractions for a given count per million (CPM) threshold are calculated over all samples (n = 3) in that group. Expression values are indicated as normalized CPM. AHARIBO-rC was performed on the ribosome pellet after sucrose cushioning. (**B**) Venn diagram of the number of lncRNAs genes with at least 1 CPM identified by RNA-seq, AHARIBO-rC, RIBO-seq, and AHARIBO RIBO-seq. (**C**) Classification of lncRNAs interacting with ribosomes and relative detection through the multiple AHARIBO and standard approaches. ND: no detection of protein synthesis. (**D**) (Left) Schematic representation of the number of mouse embryonic stem cell (mESC) lncRNAs in common between AHARIBO RIBO-seq, AHARIBO-rC RNA, and standard RIBO-seq. These lnRNAs were validated by liquid chromatography-mass spectrometry (LC-MS). (Right) Example of an AHARIBO RIBO-seq ribosome occupancy profile of lncRNA 1810058I24Rik displaying the reads distribution along the entire transcript and the accumulation of reads at the known short open reading frame (shadow area and blue arrow on top).

The online version of this article includes the following source data and figure supplement(s) for figure 4:

**Source data 1.** A table with the list of long non-coding RNAs (lncRNAs) identified by RNA-seq by RNA-seq in mouse embryonic stem cells (mESCs).

**Source data 2.** A table with the list of long non-coding RNAs (lncRNAs) identified by RIBO-seq in mouse embryonic stem cells (mESCs).

**Source data 3.** A table with the list of matching peptides from AHA-mediated RIBOsome isolation's (AHARIBO) identified long non-coding RNAs (lncRNAs).

**Figure supplement 1.** Isolation of long non-coding RNAs (lncRNAs) with AHA-mediated RIBOsome isolation (AHARIBO).

*Figure 4 continued on next page*

*Figure 4 continued*

**Figure supplement 2.** AHA-mediated RIBOsome isolation (AHARIBO) RIBO-seq data.
**Figure supplement 3.** Translated long non-coding RNAs (lncRNAs).

efficiency in detecting an ectopically expressed micropeptide (TUG1-BOAT) and (2) concordance with recently published data, prove that our approach could be useful to unravel translation events in lncRNAs that are misannotated as non-coding. Altogether, our data confirm that our three diverse and complementary AHARIBO approaches represent a unique method to identify ribosome-associated and translated RNAs.

## Discussion

LncRNAs localize in the nucleus or in the cytoplasm. In the nucleus, they modulate transcription, pre-mRNA splicing, or act as scaffold for protein interaction during chromatin organization (*Sun et al., 2018*). In the cytoplasm, the majority of lncRNAs is associated with polysomes (*Carlevaro-Fita et al., 2016*), where they either can or cannot produce proteins (*Chen et al., 2020*; *Ingolia et al., 2011*). Numerous lncRNAs are misannotated as non-coding but contain short ORFs encoding for micropeptides with biological relevance in cancer (*D'Lima et al., 2017*; *Huang et al., 2017*), bone development (*Galindo et al., 2007*), immunity (*van Solingen et al., 2018*), metabolism (*Magny et al., 2013*; *Nelson et al., 2016*), and DNA repair (*Slavoff et al., 2014*). Different methodological approaches have been developed to quantify the variations of RNA abundance by sequencing or imaging techniques (*Blumberg et al., 2019*; *Jao and Salic, 2008*; *Morisaki et al., 2016*; *Wu et al., 2016*), RNA engagement with the translational machinery by RIBO-seq or polysomal profiling (*Arava et al., 2003*; *Clamer et al., 2018*; *Eden et al., 2011*; *Taniguchi et al., 2010*), and protein synthesis by mass spectrometry or metabolic labeling (*Aviner et al., 2013*; *Dieterich et al., 2006*; *Schwanhäusser et al., 2009*; *Yan et al., 2016*). Despite these advantages, available technologies hardly capture in a single experiment the dynamics of translation across multiple biological conditions, the translation of unannotated coding transcripts, and translation-related functions of lncRNAs. Now that it is widely accepted that a portion of the genome annotated as non-coding can result in a complex transcriptome partially engaged with ribosomes (*Chen et al., 2020*; *Djebali et al., 2012*; *Iyer et al., 2015*), RNA sequencing and ribosome profiling should include micropeptide detection.

Our data show that AHARIBO serves as a flexible tool to detect translated RNAs, identify lncRNAs bound to elongating ribosomes, and detect de novo synthesized proteins. The intersection of standard RIBO-seq, RNA-seq, and AHARIBO approaches allowed us to identify translated lncRNAs. We demonstrated that AHARIBO is efficient in capturing short translated open reading frames, both native or ectopically expressed. Although LC-MS technologies are not as sensitive as RNA sequencing, we successfully identified a mouse-specific micropeptide reported to originate from a native lncRNA ORF, confirming the effectiveness of AHARIBO. To overcome existing limitations in LC-MS detection, many other translation events on lncRNAs can be predicted combining AHARIBO approaches with in silico translation of the identified leads. This approach would likely allow to selectively validate a list of still uncharacterized lncRNAs. Although the unlabeled background cannot be avoided, a pre-cleaning of the cell lysate with a cushioning step can help to increase the resolution with difficult samples. Moreover, a puromycin treatment instead of sBlock could be added as control in proteomic experiments. A unique feature of AHARIBO is the possibility to simultaneously isolate ribosomes, RNA engaged with ribosomes, and the corresponding proteins produced. Besides the versatility of the method, AHA labeling has the advantage of minimal interference with protein synthesis (*Hodas et al., 2012*; *Tom Dieck et al., 2012*).

The most prominent limitation of the method relies on the methionine starvation required for efficient AHA incorporation (*Calve et al., 2016*; *Hodas et al., 2012*; *Saleh et al., 2019*). This step can modify the physiological conditions of the cell and needs to be taken into consideration when planning experiments requiring certain stimuli (e.g., drug treatment) during methionine depletion. The conditions used in the AHARIBO protocol give robust protein labeling, but AHA concentration can be conveniently tuned based on specific cell types or biological questions. Additionally, we observed

that there are still challenges for LC-MS verification of putative lncRNA peptides identified with AHARIBO. Of note, a potential contribution from background signal needs to be taken into consideration in LC-MS and Ribo-seq analysis.

With AHARIBO we introduce a strategy for the selective isolation of active ribosomes using the nascent peptide chain as bait for a more comprehensive interrogation of lncRNA biology and proteogenomic studies. Overall, we provide evidence that AHARIBO is a comprehensive and reliable toolkit suitable for downstream parallel RNA-seq, RIBO-seq, and LC-MS analysis, empowering scientists to shed light on the functional complexity of translation.

# Materials and methods

## Key resources table

| Reagent type (species) or resource | Designation | Source or reference | Identifiers | Additional information |
|---|---|---|---|---|
| Cell line (*Homo sapiens*) | Papillomavirus-related endocervical adenocarcinoma | ATCC | RRID:CVCL_0030 | |
| Cell line (*Mus musculus*) | 46C embryonic stem cells | ATCC | RRID:CVCL_Y482 | Quattrone A. Lab. (CIBIO) |
| Antibody | Anti-β3-tubulin (mouse monoclonal) | Promega | Cat. #G712A RRID:AB_430874 | (1:2000) |
| Antibody | Anti-Oct4 (mouse monoclonal) | Santa Cruz Biotechnologies | Cat. #SC 5279 RRID:AB_628051 | (1:2000) |
| Antibody | Anti-human RPL26 (rabbit polyclonal) | Abcam | Cat. #ab59567 RRID:AB_945306 | (1:2000) |
| Antibody | Anti-human RPS6 (rabbit polyclonal) | Abcam | Cat. #ab40820 RRID:AB_945319 | (1:2000) |
| Antibody | Anti-human beta actin (rabbit polyclonal) | Abcam | Cat. #ab8227 RRID:AB_2305186 | (1:2000) |
| Recombinant DNA reagent | WT TUG1-BOAT (plasmid) | PMID:32894169 | | |
| Recombinant DNA reagent | Δ TUG1-BOAT (plasmid) | This paper | | See 'Materials and methods section: 'TUG1-BOAT ectopic expression and qPCR' |
| Recombinant DNA reagent | +1Met TUG1-BOAT (plasmid) | This paper | | See 'Materials and methods' section: 'TUG1-BOAT ectopic expression and qPCR' |
| Peptide, recombinant protein | Precision Protein StrepTactin-HRP Conjugate | BioRad | Cat. #1610380 | (1:5000) |
| Chemical compound, drug | L-Arginine-13C6,15N4 hydrochloride | Sigma-Aldrich | Cat. #608033 | |
| Chemical compound, drug | L-Lysine-13C6,15N2 hydrochloride | Sigma-Aldrich | Cat. #608041 | |
| Chemical compound, drug | L-Azidohomoalanine (Click-IT AHA) | Invitrogen | Cat. #C10102 | |
| Chemical compound, drug | Dibenzocyclooctyne-PEG4-biotin conjugate | Sigma-Aldrich | Cat. #760749SML1656 | |
| Chemical compound, drug | sBlock | IMMAGINA BioTechnology | Cat. #SM8 | |
| Chemical compound, drug | Puromycin | Sigma-Aldrich | Cat. #P8833 | |
| Chemical compound, drug | Cycloheximide | Sigma-Aldrich | #C4859 | |
| Chemical compound, drug | Lipofectamine 3000 Transfection Reagent | Thermo Fisher Scientific. | Cat. #L3000001 | |
| Chemical compound, drug | Mag-DBCO beads | IMMAGINA BioTechnology | Cat. #MDBCO | |

*Continued on next page*

*Continued*

| Reagent type (species) or resource | Designation | Source or reference | Identifiers | Additional information |
|---|---|---|---|---|
| Chemical compound, drug | eMagSi-cN beads | IMMAGINA BioTechnology | #018-eMS-001 | |
| commercial assay or kit | SMART-Seq Stranded Kit | Takara | Cat. #634443 | |
| Commercial assay or kit | SuperScript III Reverse Transcriptase | Thermo Fisher | Cat. #18080044 | |
| Commercial assay or kit | Kapa Probe Fast Universal qPCR Kit | Kapa Biosystems | #KK4702 | |
| Software, algorithm | Image analysis | ImageJ | RRID:SCR_003070 | |
| Software, algorithm | Statistical package | edgeR | RRID:SCR_012802 | |

## Cell culturing and treatments

For protocol development, optimization, and validation, HeLa cells were used. HeLa cells were maintained on adherent plates in Dulbecco's modified Eagle's medium (DMEM; EuroClone #ECM0728L) supplemented with 10% fetal bovine serum, 2 mM L-glutamine, 100 units/mL penicillin, and 100 μg/mL streptomycin at 37°C, 5% $CO_2$. For passaging, cells were washed with 1× Phosphate-Buffered Saline (PBS), detached using 0.25% trypsin-EDTA, and spun down at 260 × *g* for 5 min.

For treatments, 250,000–400,000 HeLa cells per well were seeded in six-well plates and grown to 80% confluence. At the time of treatment, culture medium was removed and cells were washed once with warm 1× PBS. Subsequently, cells were incubated with Dulbecco's modified Eagle's limiting medium (DMEM-LM; Thermo Scientific #30030) supplemented with 10% fetal bovine serum and 800 μM L-leucine for 40 min to deplete methionine reserves. Methionine-free medium was then supplemented with L-azidohomoalanine (Click-IT AHA; Invitrogen #C10102) at a final concentration of 250 μM and incubation time (ranging from 10 min to 120 min; 30 min set as incubation time for the protocol). Cells were then treated with 1× sBlock (IMMAGINA BioTechnology, catalog no. #RM8; sBlock is an anisomycin-containing proprietary reagent) for 10 min. Then, six-well plates were placed on ice, medium was removed, and cells were washed once with cold 1× PBS supplemented with 1× sBlock. After removing residual PBS with a pipette, hypotonic lysis buffer (0.01 M NaCl, 0.01 M $MgCl_2$, 0.01 M Tris-HCl, 1% Tx-100, 1× sBlock, 1% sodium deoxycholate, 5 units/mL DNAse I [Thermo Scientific #89836], 200 units/mL RiboLock RNase Inhibitor [Thermo Scientific #EO0381], 1× Protease Inhibitor Cocktail [Cell Signaling Technology #5871S]) was added to each well, and cells were lysed with the aid of a scraper. After hypotonic lysis, nuclei and cellular debris were removed by centrifuging at 18,000 × *g*, 4°C for 5 min. For quantification of the total absorbance value of cell lysates, the absorbance was measured (260 nm) using a Nanodrop ND1000 UV-VIS Spectrophotometer. Lysates were aliquoted and processed directly or stored at −80°C.

Arsenite pre-treatment was performed by adding sodium arsenite (Sigma-Aldrich #S7400) at a final concentration of 500 μM for 1 hr.

For RNA-seq and proteomics experiments, two biological settings were assessed in triplicate experiments: (1) undifferentiated mouse 46C embryonic stem cells (mESCs) (*Ying et al., 2003*) and (2) mESCs induced to differentiate into ENs. mESCs were maintained in mESC self-renewal medium composed of Glasgow's MEM (Thermo Scientific #11710-035) supplemented with 1000 units/mL ESGRO Recombinant Mouse LIF protein (Millipore #ESG1107), 10% fetal bovine serum, 55 μM 2-mercaptoethanol, 1 mM sodium pyruvate (Thermo Scientific #11360070), MEM non-essential amino acids (Thermo Scientific #11140050), GlutaMax (Thermo Scientific #35050061), and penicillin/streptomycin. For passaging, mESCs were washed twice with 1× PBS, detached using 0.02–0.05% trypsin-EDTA, and spun down at 260 × *g* for 3 min. Pellet was resuspended in fresh medium and plated onto 0.1% gelatin-coated culture vessels.

For treatments, $5 \times 10^5$ mESCs/$cm^2$ were seeded in Petri dishes and grown to 60% confluence. For pSILAC proteomics, 24 hr before lysis mESCs were washed twice with 1× PBS and the medium was replaced with SILAC Advanced DMEM/F-12 Flex Medium (Thermo Scientific #A2494301), supplemented with 1000 units/mL ESGRO Recombinant Mouse LIF protein, 10% dialyzed fetal bovine

serum, 4500 mg/L glucose, 17.25 mg/L proline, and penicillin/streptomycin. Either light or heavy L-arginine (Sigma-Aldrich #608033) and L-lysine (Sigma-Aldrich #608041) were added at 84 mg/L and 146 mg/L, respectively. For both AHA+ proteomics and RNA-seq experiments, treatments were performed as described above for HeLa cells, with the exception that methionine-free medium was supplemented with 1000 units/mL ESGRO Recombinant Mouse LIF protein and 10% dialyzed fetal bovine serum. After methionine depletion, cells were treated with 250 µM AHA for 30 min. The remaining treatment steps and hypotonic lysis were performed as detailed above.

Neuronal differentiation was performed according to a previously described protocol (*Ying et al., 2003*). Briefly, 2.000 mESCs/cm$^2$ were seeded on gelatin-coated culture vessels in N2B27 medium. Cells were gently washed with 1× PBS, and medium was renewed every 1–2 days until 15DIV. N2B27 medium is composed of 1:1 mix of DMEM/F-12 (Thermo Scientific #21331020) and Neurobasal Medium (Thermo Scientific #21103049), supplemented with 0.5% N-2 (Thermo Scientific #17502048), 1% B-27 (Thermo Scientific #17504044), GlutaMax, and penicillin/streptomycin.

Upon differentiation, ENs were treated directly in culture vessels. For pSILAC proteomics, 24 hr before lysis ENs were washed once with 1× PBS and the medium was replaced with SILAC Advanced DMEM/F-12 Flex Medium, supplemented with 0.5% N2, 1% B27, 4500 mg/L glucose, 17.25 mg/L proline, and penicillin/streptomycin, 4500 mg/L glucose, 17.25 mg/L proline, and penicillin/streptomycin. Either light or heavy L-arginine and L-lysine were added at 84 mg/L and 146 mg/L, respectively. For both AHA+ proteomics and RNA-seq experiments, ENs were treated as described above for HeLa cells, with 250 µM AHA for 30 min. The remaining treatment steps and hypotonic lysis were performed as detailed above.

Cell lines were purchased directly from ATCC and passaged fewer than 15 times. *Mus musculus* 46C ES were obtained from Quattrone A. Lab (CIBIO, RRID:CVCL_Y482). All cells tested negative for mycoplasma contamination.

## Immunocytochemistry

For immunofluorescence assay, cells were fixed with 4% paraformaldehyde for 15 min at room temperature, permeabilized using 0.5% Triton X-100 in 1× PBS for 15 min at room temperature, and blocked using 5% fetal bovine serum, 0.3% Triton X-100 in 1× PBS for 2 hr at room temperature. Cultures were then incubated overnight at 4°C with either anti-β3-tubulin (Promega #G712A) or anti-Oct4 (Santa Cruz Biotechnologies #SC-5279) primary antibodies diluted in 2% fetal bovine serum, 0.2% Triton X-100 in 1× PBS. Cells were then washed three times with 1× PBS and incubated with Alexa-555 anti-mouse secondary antibodies for 2 hr. Nuclei were counterstained with Hoechst 33258 before imaging with a Zeiss Axio Observer Z1 inverted microscope equipped with a 2.83 Megapixel AxioCam 503 mono D camera.

## AHARIBO-rC/AHARIBO-nP: purification of active ribosomes for RNA/ protein isolation

For RNA-seq experiments, lysates were diluted in W-buffer (10 mM NaCl, 10 mM MgCl$_2$, 10 mM HEPES, 1× sBlock) to a final Nanodrop-measured absorbance (260 nm) of 1–2 a.u./mL, supplemented with 40 U of Superase-In RNase Inhibitor (Thermo Scientific #AM2696) and incubated with dibenzocyclooctyne-PEG4-biotin conjugate (Sigma-Aldrich #760749; 50 µM final concentration) in a reaction volume of 100 µL for 1 hr on a rotator in slow motion (9 rpm) at 4°C. Lysates were then incubated with 50 µL of eMagSi-cN beads (IMMAGINA BioTechnology #018-eMS-001) for 30 min at 4°C on the rotator in slow motion (9 rpm). Subsequently, samples were taken off the rotator and placed on a magnetic rack on ice, and supernatants were discarded. Beads were washed two times with 500 µL of 1× PBS supplemented with 0.1% Triton-X100, 1× sBlock, and 1:10,000 RiboLock RNase Inhibitor (Thermo Scientific #EO0381) on the rotator in slow motion at 4°C, removing supernatants from the tubes sitting on the magnetic rack and gently adding new washing solution each time. After the final wash, beads were resuspended in 200 µL of W-buffer and transferred to a new vial. Then, 20 µL of 10% SDS and 5 µL of Proteinase K (Qiagen #19131) were added to each sample, and samples were incubated at 37°C for 75 min in a water bath. Subsequently, suspensions were transferred to a new vial, and acid phenol:chloroform:isoamyl alcohol RNA extraction was performed. Briefly, an equal volume of acid phenol:chloroform:isoamyl alcohol (pH 4.5) was added, and samples were vortexed and centrifuged at 14,000 × *g* for 5 min. Aqueous phases were then transferred to new vials,

500 µL of isopropanol and 2 µL of GlycoBlue (Thermo Scientific #AM9516) were added, samples were mixed and incubated at room temperature for 3 min, and then stored overnight at −80℃. The following day samples were centrifuged at 14,000 × g for 30 min, supernatants were removed, 500 µL of 70% ethanol were added to each sample, and samples were then centrifuged at 14,000 × g for 10 min. Finally, pellets were air-dried and resuspended in 10 µL of nuclease-free water. When quality check and quantification was needed, RNA samples were run on a 2100 Bioanalyzer (Agilent) using the Agilent RNA 6000 Nano Reagents kit (Agilent #5067–1511) and assayed on the Qubit fluorometer using the Qubit RNA HS Assay Kit (Thermo Scientific #Q32852). For visualization of total RNA patterns, samples were run on a 1% agarose gel. ImageJ software (v 1.45s) was used for the quantitation of signal intensities of ribosomal RNA bands.

For proteomics experiments, lysates were diluted in W-buffer to a final Nanodrop-measured absorbance (260 nm) of 1–2 a.u./mL in a final volume of 100 µL. Ribosome pulldown was performed using Mag-DBCO beads (IMMAGINA BioTechnology #MDBCO). Lysates were incubated with 50 µL of beads for 1 hr on a rotator in slow motion (9 rpm) at 4℃. Supernatants were discarded after placing samples on the magnetic rack. Beads were washed three times with 500 µL of 200 mM Tris, 4% CHAPS, 1 M NaCl, 8 M urea, and pH 8.0 at room temperature on a shaker at 1000 rpm, using the magnetic rack to replace the washing solution. After the final wash, beads were resuspended in 30 µL of water and transferred to a new vial.

## qRT-PCR analysis

Total RNA was extracted from samples processed through the AHARIBO-rC protocol as described above. Depending on the available input material, RNA was retrotranscribed using either RevertAid First Strand cDNA Synthesis Kit (Thermo Scientific #K1621) or SuperScript III Reverse Transcriptase (Thermo Fisher #18080044), per manufacturer's protocols. qPCR was run on CFX Connect Real-Time PCR Detection System (BioRad) using Kapa Probe Fast Universal qPCR Kit (Kapa Biosystems #KK4702). Reactions were performed in technical duplicates of biological triplicates. The following TaqMan probes were used: Hs99999901_s1 (18S), Hs02800695_m1 (HPRT1).

For the normalization of qRT-PCR results, HPRT1 was used as housekeeping gene. The fold change in normalized 18S RNA levels between untreated (control) and treated (arsenite) samples was calculated. A second normalization to threshold cycles from non-AHA-treated samples was done to account for background signal.

## TUG1-BOAT ectopic expression and qPCR

We ectopically express the putative protein produced by the open reading frame of TUG1, called TUG1-BOAT (Tug1-Bifunctional ORF and Transcript), in HeLa cells. Briefly, construct generation and transfection was performed as in *Lewandowski et al., 2020* with some minor changes to adapt the experimental setup to the AHARIBO method. We synthesized three different constructs for human Tug1 ORF1 (Thermo Scientific):

1. The first (called *WT TUG1-BOAT*) is the one reported in *Lewandowski et al., 2020*. It has a non-canonical start codon and a methionine at 75 nt (25 aa) upstream of the stop codon.
2. The second (called Δ TUG1-BOAT) is deleted by the only methionine of the sequence present at 75 nt from the stop codon. No methionines are present.
3. The third (called +1Met TUG1-BOAT) has an ATG start codon (methionine) instead of the non-canonical CTG start codon e. Therefore, the third construct has two methionines, one at the N terminal and the other at 25 aa (about 75 nt) upstream of the C-terminal.

We cloned the constructs in the pcDNA3.1(+) plasmid with *Hind*III and *Eco*RV restriction enzymes. For transfection of TUG1-BOAT constructs, we seeded HeLa cells in a six-well plate and transfected the cells with 2.5 µg of plasmids (pcDNA3.1(+) containing each of the inserts) using 742 Lipofectamine 3000 Transfection Reagent (Thermo Fisher Scientific). After 24 or 48 hr post transfection, cells were processed with AHARIBO-rC protocol followed by RNA extraction.

We performed qPCR analysis on AHARIBO pulldowns and input for each vector to validate the efficiency in capturing short translated ORF deriving from RNA annotated as lncRNA (TUG1). Briefly, 200 ng of DNase I-treated RNA was used as input to generate cDNA using High-Capacity cDNA Reverse Transcription Kit (Applied Biosystems), according to the manufacturer's protocol. qPCR was run on CFX Connect Real-Time PCR Detection System (BioRad) using Powerup Sybr Master Mix

(Applied Biosystems) and a couple of primers design to amplify 150 nt of the CDS of TUG1-BOAT transcript (see below). Reactions were performed in technical duplicates of biological duplicates. For normalization, 18S was used as housekeeping gene. Ct values were analyzed using the ΔΔCt method (*Livak and Schmittgen, 2001*).

> Fw PRIMER: GGCTCTTTCTCCTGCTCTGG
> Rev PRIMER: CTCCTCGTCGAATCGCAAAC
> Insert size: 150 nt

TUG1-BOAT sequences are listed below:

Italics: 5′ UTR leader sequence; bold: canonical and non-canonical start codons; red: methionine.

>WT TUG1-BOAT
*GGCCGAGCGACGCAGCCGGGACGGTAGCTGCGGTGCGGACCGGAGGAGCCATCTTGTC*
*TCGTCGCCGGGGAGTCAGCCCCTAAATCGAAGAAGCC***CTG**GCGCGCCCTCCCCCCC
TCCCGGGTCTGGTAGGGCGAAGGAACGGGCGTGCGGTCGATCGAGCGATCGG
TTGGCGGCTCTTTCTCCTGCTCTGGCATCCAGCTC
TTGGGGCGCAGGCCCGGCCGCCGCGGCGCGCGCCCGGTGGCCGTTGGCGC
TCGCGCCGCGTCTTTCTTCTCGTACGCAGAACTCGGGCGGCGGCCTATGCGTTTGCGA
TTCGACGAGGAGTCGTCCGGGTGGTCGGCGGCGGCGGGCAGCTGCTCCGCCCCGC
TCCGGGGGGAGGCGGCGGCGGCAGCGGCCGCGGGATTTGGAGCGGCCGGG-
GAGGCGGGGGTGGCCGGGGCCGGCTTGGAGGCCTGGCGCCACCCTTCGGGGCC
TGCAAGGACCCAGTTGGGGGGGCAGGAGGGGGCCGGAGGATGGTTGGTTGTGGGATTTC
TACTTTGCCTTTTCCTCCTTATGCCGCCTGACTACAAAGACCATGACGGTGATTATAAAGA
TCATGACATCGACTACAAGGATGACGATGACAAGTAG

>ΔTUG1-BOAT
*GGCCGAGCGACGCAGCCGGGACGGTAGCTGCGGTGCGGACCGGAGGAGCCATCTTGTC*
*TCGTCGCCGGGGAGTCAGCCCCTAAATCGAAGAAGCC***CTG**GCGCGCCCTCCCCCCC
TCCCGGGTCTGGTAGGGCGAAGGAACGGGCGTGCGGTCGATCGAGCGATCGG
TTGGCGGCTCTTTCTCCTGCTCTGGCATCCAGCTC
TTGGGGCGCAGGCCCGGCCGCCGCGGCGCGCGCCCGGTGGCCGTTGGCGC
TCGCGCCGCGTCTTTCTTCTCGTACGCAGAACTCGGGCGGCGGCCTATGCGTTTGCGA
TTCGACGAGGAGTCGTCCGGGTGGTCGGCGGCGGCGGGCAGCTGCTCCGCCCCGC
TCCGGGGGGAGGCGGCGGCGGCAGCGGCCGCGGGATTTGGAGCGGCCGGG-
GAGGCGGGGGTGGCCGGGGCCGGCTTGGAGGCCTGGCGCCACCCTTCGGGGCC
TGCAAGGACCCAGTTGGGGGGGCAGGAGGGGGCCGGAGGATGGTTGGTTGTGGGATTTC
TACTTTGCCTTTTCCTCCTTCCGCCTGACTACAAAGACCATGACGGTGATTATAAAGATCA
TGACATCGACTACAAGGATGACGATGACAAGTAG

>+1Met TUG1-BOAT
*GGCCGAGCGACGCAGCCGGGACGGTAGCTGCGGTGCGGACCGGAGGAGCCATCTTGTC*
*TCGTCGCCGGGGAGTCAGGCCCCTAAATCGAAGAAGCC***ATG**GACTACAAGGATGACGA
TGACAAGGCGCGCCCTCCCCCCCTCCCGGGTCTGGTAGGGCGAAGGAACGGGCGTGCGG
TCGATCGAGCGATCGGTTGGCGGCTCTTTCTCCTGCTCTGGCATCCAGCTC
TTGGGGCGCAGGCCCGGCCGCCGCGGCGCGCGCCCGGTGGCCGTTGGCGC
TCGCGCCGCGTCTTTCTTCTCGTACGCAGAACTCGGGCGGCGGCCTATGCGTTTGCGA
TTCGACGAGGAGTCGTCCGGGTGGTCGGCGGCGGCGGGCAGCTGCTCCGCCCCGC
TCCGGGGGGAGGCGGCGGCGGCAGCGGCCGCGGGATTTGGAGCGGCCGGG-
GAGGCGGGGGTGGCCGGGGCCGGCTTGGAGGCCTGGCGCCACCCTTCGGGGCC
TGCAAGGACCCAGTTGGGGGGGCAGGAGGGGGCCGGAGGATGGTTGGTTGTGGGATTTC
TACTTTGCCTTTTCCTCCTTATGCCGCCTGACTACAAAGACCATGACGGTGATTATAAAGA
TCATGACATCTAG

## RNA-seq

RNA samples were subjected to library preparation for the Illumina platform using the SMART-Seq Stranded Kit (Takara #634443) as per manufacturer's instructions using 5 ng of RNA as starting material. For quality check and quantification, the final libraries were run on a 2100 Bioanalyzer (Agilent) using the Agilent DNA 1000 Kit (Agilent #5067-1504) and assayed on the Qubit fluorometer using the Qubit dsDNA HS Assay Kit (Thermo Scientific #Q32851). Libraries were sequenced on an Illumina HiSeq2500 by the NGS Core Facility (University of Trento).

## Polysome profiling

HeLa cells were treated and lysed as described above, adding one of the following blocking drugs: (1) sBlock (IMMAGINA BioTechnology #SM8, final concentration 1×, 10 min treatment); (2) cycloheximide (Sigma-Aldrich #C4859; final concentration 30 µM, 5 min treatment); (3) puromycin (Sigma-Aldrich #P8833; final concentration 50 µM, 5 min treatment); and (4) no blocking drug. Cleared supernatants obtained from cytoplasmic lysates were loaded on a linear 15–50% sucrose gradient and ultracentrifuged in a SW41Ti rotor (Beckman) for 1 hr and 40 min at 180,000 × $g$ at 4°C in a Beckman Optima LE-80K Ultracentrifuge. After ultracentrifugation, gradients were fractionated in 1 mL volume fractions with continuous absorbance monitoring at 254 nm using an ISCO UA-6 UV detector. Each fraction was flash-frozen in liquid nitrogen and stored at −80°C for subsequent protein extraction.

Polysome profiles were analyzed as follows. The relative intensity of each individual fraction was determined for both on-membrane AHA and RPL26 signals, then the AHA/RPL26 relative intensity ratio was calculated for each fraction. For each profile, the relative intensity ratios of polysome-containing fractions (fractions 8/9–10/11) were averaged and normalized to the relative intensity ratio of the 60S fraction, which was chosen as internal baseline for background signal based on the fact that it should be devoid of translationally active ribosomes. To assess the effect of the different blocking drugs, averaged normalized relative intensity ratios for the profiles obtained from different blocking drugs and from the untreated control sample were compared. ImageJ software (v 1.45s) was used for quantitation of signal intensities of protein bands.

## Sucrose cushioning for ribosome enrichment (global translatome)

HeLa cells were treated in Petri dishes and lysed as described above, adding 1× sBlock as blocking drug. Sucrose cushioning was performed according to a modified version of a previously described protocol (*Ingolia et al., 2012*). For each sample, a volume of cell lysate corresponding to 1.7 a.u. (based on Nanodrop measurement of absorbance at 260 nm) was layered on top of 900 µL of 30% sucrose cushion (30 mM Tris-HCl pH 7.5, 100 mM NaCl, 10 mM MgCl$_2$, 1 M sucrose in nuclease-free water) supplemented with 1× sBlock. Samples were ultracentrifuged at 95,000 rpm at 4°C for 1 hr and 40 min using a TLA$_{100.2}$ rotor (Beckman). Pellets were resuspended in 100 µL of nuclease-free water supplemented with 30 mM Tris-HCl, pH 7.5, 100 mM NaCl, 10 mM MgCl$_2$.

## Protein extraction from sucrose gradient fractions

Polysomal fractions (1 mL) or pellet/supernatant fractions from 30% sucrose cushioning (1/5th of total amount, adjusted to 260 µL volume) were processed for methanol/chloroform protein extraction. Briefly, 600 µL of methanol and 150 µL of chloroform were added to each sample and samples were vortexed. Then, 450 µL of deionized water were added to each sample and samples were vortexed again. Samples were centrifuged at 14,000 × $g$ for 1 min at room temperature, and the resulting aqueous phase was removed without disrupting the underlying white ring (protein interface). Subsequently, 450 µL of methanol were added to each sample, samples were vortexed, and then centrifuged at 14,000 × $g$ for 2 min at room temperature. After centrifugation, supernatants were removed and pellets air-dried. Finally, pellets were resuspended in deionized water supplemented with Pierce Lane Marker Reducing Sample Buffer (Thermo Scientific #39000) to a final volume of 15 µL and either stored at −80°C or heated at 95°C and directly used for SDS-PAGE.

## On-membrane click chemistry

Cell lysate or protein extracts obtained from sucrose gradient fractions were supplemented with Pierce Lane Marker Reducing Sample Buffer (Thermo Scientific #39000), heated at 95°C for 10 min, and separated by SDS-PAGE. Separated proteins were transferred to nitrocellulose membranes, then membranes were blocked overnight at 4°C in 5% milk prepared in 1× Tris-Buffered Saline (TBS) − 0.1% Tween20 supplemented with dibenzocyclooctyne-PEG4-biotin conjugate (Sigma-Aldrich #760749; 50 µM final concentration). Membranes were washed three times in 1× TBS − 0.1% Tween20 for 10 min each, then incubated with Precision Protein StrepTactin-HRP Conjugate (BioRad #1610380; 1:1000 in 5% milk prepared in 1× TBS − 0.1% Tween20) for 1 hr at room temperature, then washed again. Membranes were subsequently developed using Amersham ECL Prime Western Blotting Detection Reagent (GE Healthcare #RPN2236). Images were acquired

through the ChemiDoc MP Imaging System. ImageJ software (v 1.45s) was used for quantitation of AHA signal intensities.

## Immunoblotting

Aliquots of 10–20 µL of cell lysate or protein extracts obtained from sucrose gradient fractions were supplemented with Pierce Lane Marker Reducing Sample Buffer (Thermo Scientific #39000), heated at 95℃ for 10 min, and separated by SDS-PAGE. Separated proteins were transferred to nitrocellulose membranes, then membranes were blocked for 1 hr at room temperature in 5% milk prepared in 1× TBS − 0.1% Tween20. Membranes were subsequently incubated for 1 hr at room temperature with the following primary antibodies, diluted in 5% milk prepared in 1× TBS − 0.1% Tween20: anti-RPL26 (Abcam #ab59567; 1:2000), anti-RPS6 (Abcam #ab40820; 1:1000), and anti-beta-actin (Abcam #ab8227; 1:2000). Membranes were washed three times in 1× TBS − 0.1% Tween20 for 10 min each, then incubated with the appropriate HRP-conjugated secondary antibodies for 1 hr at room temperature and washed again as before. Membranes were then developed using either Amersham ECL Prime Western Blotting Detection Reagent (GE Healthcare #RPN2236) or SuperSignal West Femto Maximum Sensitivity Substrate (Thermo Scientific #34095), depending on signal intensities. Images were acquired through the ChemiDoc MP Imaging System. ImageJ software (v 1.45s) was used for quantitation of signal intensities of protein bands.

## RNA-seq data analysis

FASTQ files from Illumina HiSeq2500 were first checked for adapters and quality-base distribution using FASTQC tool (http://www.bioinformatics.babraham.ac.uk/projects/fastqc), followed by trimming with Trimmomatic-0.35 (*Bolger et al., 2014*) with the following settings: ILLUMINACLIP:ADAPTOR_FILE:2:30:8:1 LEADING:3 TRAILING:3 SLIDINGWINDOW:4:15. Prior to sequencing data processing, technical replicates (different sequencing lanes) from the same library were merged together generating a unique FASTQ per sample. Reads were aligned onto mm10 Mouse genome using STAR-2.6.0a (*Dobin et al., 2013*) with a maximum mismatch of two and default setting for all other parameters. Once uniquely mapped reads were selected, the GRCm38.92 mouse gene annotation from Ensembl (http://www.ensembl.org) was incorporated in the HTSeq-count v0.5.4 (*Anders et al., 2015*) tool to obtain gene-level counts. Genes with counts per million (CPM) <1 in all samples were considered as not expressed and hence removed from the analysis. Trimmed mean of M values (TMM) (*Robinson and Oshlack, 2010*) normalization and CPM conversion were then performed to obtain gene expression levels for downstream analyses. For each comparison, differential expression testing was performed using the edgeR-3.20.9 (*Robinson et al., 2010*) statistical package from Bioconductor. According to the edgeR approach, both common (all genes in all samples) and separate (gene-wise) dispersions were estimated and integrated into a negative binomial generalized linear model to moderate gene variability across samples. Finally, genes having a Log fold change higher/smaller than 1.5/−1.5 and an FDR-corrected p-value of 0.01 (or smaller) were considered as significant.

## Proteomics experiments

Proteomic analysis was performed on samples processed through the pSILAC and AHARIBO workflows, as described above. For pSILAC experiments, cells were prepared as described above (see 'Cell culturing and treatments'). Then, 50 µg of lysates was subjected to acetone precipitation and protein pellets were dissolved in 50 mM ammonium bicarbonate and 6 M urea. For AHARIBO enrichment, the beads used for ribosome pulldown were reconstituted in 100 µL 6 M urea with 50 mM ammonium bicarbonate.

Samples were reduced using 10 mM DTT, 1,4-Dithiothreitol (DTT) for 1 hr at room temperature and alkylated with 20 mM iodoacetamide in the dark for 30 min at room temperature. Subsequently, proteins were digested at room temperature with 0.5 µg Lys-C (Promega, #VA1170) for 4 hr, after which the solution was diluted four times in 50 mM ammonium bicarbonate. Then, 1 µg of trypsin (Promega, #V5111) was added to the samples and proteolysis was carried out overnight. Digestion was interrupted by adding 1% trifluoroacetic acid. Samples were then desalted by C18 stage-tip, lyophilized, and resuspended in 20 µL of buffer A (0.1% formic acid) for LC-MS/MS analysis.

Samples were analyzed using an Easy-nLC 1200 system coupled online with an Orbitrap Fusion Tribrid mass spectrometer (both Thermo Fisher Scientific). Peptides were loaded onto a 25-cm-long Acclaim PepMap RSLC C18 column (Thermo Fisher Scientific, 2 µm particle size, 100 Å pore size, id 75 µm) heated at 40°C. For pSILAC samples, the gradient for peptide elution was set as follows: 5–25% buffer-B (80% acetonitrile, 0.1% formic acid) over 90 min, 25–40% over 15 min, 40–100% over 18 min, and 100% for 17 min at a flow rate of 400 nL/min. For AHARIBO pulldown samples, the same steps for peptide elution were set over a total gradient of 80 min. The instrument was set in a data-dependent acquisition mode. The full MS scan was 350–1100 m/z in the orbitrap with a resolution of 120,000 (200 m/z) and an AGC target of $1 \times 10^6$. MS/MS was performed in the ion trap using the top speed mode (3 s), an AGC target of $5 \times 10^3$, and an HCD collision energy of 30.

MS raw files were analyzed by using Proteome Discoverer (v2.2, Thermo Scientific). MS/MS spectra were searched by the SEQUEST HT search engine against the human or the mouse UniProt FASTA databases (UniProtKB 11/2018). Trypsin was specified as the digestive enzyme. Cysteine carbamidomethylation (+57.021 Da) was set as fixed modification, and methionine oxidation (+15.995 Da) and N-term acetylation (+42.011 Da) as variable modifications. SILAC labeling (Lys +8.014 Da, Arg +10.008 Da) was used as quantification method for pSILAC samples. All other values were kept as default.

## Proteomics data analysis

Heteroscedastic T-test was used to assess the significant differences in peptide/protein abundance (p-value lower than 0.05) unless stated otherwise. Data distribution was assumed to be normal, but this was not formally tested. GO and Kyoto Encyclopedia of Genes and Genomes pathway analysis were performed using DAVID version 6.8, PANTHER 14.1, and Enrichr (http://amp.pharm.mssm.edu/Enrichr/).

## Identification of lncRNA peptides from result spectra

Sequenced non-coding RNAs were in silico translated into amino acid sequences using the EMBOSS Transeq tool from EMBL. Only the three forward frames were translated. Spectra obtained from the AHARIBO enrichment of newly synthesized proteins were searched against a database of typical contaminants like keratins, trypsin, and bovine serum albumin provided by MaxQuant (*Cox and Mann, 2008*). The software utilized for database searching was Proteome Discoverer (v2.4, Thermo Scientific); the non-fragment filter and the Top N Peaks Filter (with N = 4 per 100 Da) were also used in the workflow to eliminate noise signals from the MS/MS spectra. The spectra not matching with high confidence this database were searched against the human SwissProt database. Those not matching with both databases were used to match the in silico translated database generated by EMBOSS Transeq using semi-specific tryptic cleavage to consider also unexpected translation start sites. We considered only those spectra that passed the 1% FDR threshold and created two distinct groups for those peptides with an AUG 'in-frame' versus not in-frame.

### Ribosome profiling

mESCs at 80% confluence were pre-treated with the elongation inhibitor cycloheximide before rapid harvest on ice and cell lysis (lysis buffer, IMMAGINA BioTechnology #RL001-1). Clarified cell lysates (1.7 total a.u., measured by Nanodrop) were treated with 1.3 U of RNase I (Thermo, #AM2295) in W-buffer (IMMAGINA BioTechnology #RL001-4) containing 1× sBlock to digest RNA not protected by ribosomes. Digestion was performed for 45 min at RT and then stopped with Superase-In RNase Inhibitor (Thermo Scientific #AM2696) for 10 min on ice. Samples were then processed differentially according to the specific approaches described below.

### Standard RIBO-seq

80S ribosomes were isolated by centrifuging lysates through a 30% sucrose cushion at 95,000 rpm, for 2 hr at 4°C. The cushion was resuspended in W-buffer and treated with SDS 10% (final 1%) and 5 µL of proteinase K (20 mg/mL), and incubated at 37°C in a water bath for 75 min before acid phenol:chloroform:isoamyl alcohol (pH 4.5) RNA extraction.

AHARIBO RIBO-seq

The lysates were incubated with dibenzocyclooctyne-PEG$_4$-biotin conjugate (Sigma-Aldrich #760749; 50 μM final concentration) in a reaction volume of 100 μL for 1 hr on a rotator in slow motion (9 rpm) at 4˚C. Lysates were then incubated with 50 μL of eMagSi-cN beads (IMMAGINA BioTechnology #018-eMS-001) for 30 min at 4˚C on the rotator in slow motion (9 rpm). Subsequently, samples were taken off the rotator and placed on a magnetic rack on ice, and supernatants were discarded. Beads were washed two times with 500 μL of 1× PBS supplemented with 0.1% Triton-X100, 1× sBlock, and 1:10,000 RiboLock RNase Inhibitor (Thermo Scientific #EO0381) on the rotator in slow motion at 4˚C, removing supernatants from the tubes sitting on the magnetic rack and gently adding new washing solution each time. After the final wash, beads were resuspended in 200 μL of W-buffer and transferred to a new vial. Then, 20 μL of 10% SDS and 5 μL of Proteinase K (Qiagen #19131) were added to each sample, and samples were incubated at 37˚C for 75 min in a water bath. Subsequently, suspensions were transferred to a new vial, and acid phenol:chloroform:isoamyl alcohol (pH 4.5) RNA extraction was performed.

For both approaches, protocol steps starting from RNA extraction were performed as follows. Briefly, an equal volume of phenol:chloroform:isoamyl alcohol was added, and samples were vortexed and centrifuged at 14,000 × g for 5 min. Aqueous phases were then transferred to new vials, 500 μL of isopropanol and 2 μL of GlycoBlue (Thermo Scientific #AM9516) were added, samples were mixed and incubated at room temperature for 3 min, and then stored overnight at −80˚C. The following day samples were centrifuged at 14,000 × g for 30 min, supernatants were removed, 500 μL of 70% ethanol were added to each sample, and samples were then centrifuged at 14,000 × g for 10 min. Finally, pellets were air-dried and resuspended in 10 μL of nuclease-free water. Extracted RNA was then resolved by electrophoresis through denaturing TBE-urea gels, and fragments between 25 nt and 35 nt were excised. Libraries were prepared using the RiboLace kit_module 2 (IMMAGINA BioTechnolgy #RL001_mod2) and sequenced on an Illumina HiSeq 2500 sequencer with a single-end 50 bp run.

## RIBO-seq data analysis

Reads were processed by removing 5' adapters, discarding reads shorter than 20 nucleotides, and trimming the first nucleotide of the remaining ones (using Trimmomatic v0.36). Reads mapping on the collection of *M. musculus* rRNAs (from the SILVA rRNA database, release 119) and tRNAs (from the Genomic tRNA database: gtrnadb.ucsc.edu/) were removed. Remaining reads were mapped on the mouse transcriptome (using the Gencode M17 transcript annotations). Antisense and duplicate reads were removed. All alignments were performed with STAR (v020201) employing default settings.

The identification of the P-site position within the reads was performed using riboWaltz (v1.1.0) computing the P-site offsets from the 3' end of the reads. The percentage of P-sites falling in the three annotated transcript regions (5' UTR, CDS, and 3' UTR) was computed using the function *region_psite* included in riboWaltz (v1.1.0). Transcript counts were normalized using the TMM method implemented in the edgeR Bioconductor package. Transcripts displaying 1 CPM in at least one replicate were kept for further analyses.

## Additional information

### Competing interests

Luca Minati: L.M. is an employee of IMMAGINA BioTechnology S.r.l. Claudia Firrito: C.F. is an employee of IMMAGINA BioTechnology S.r.l. Alessia Del Piano: A.D.P. is an employee of IMMAGINA BioTechnology S.r.l. Alberto Peretti: A.P. is an employee of IMMAGINA BioTechnology S.r.l. Paola Bernabo: P.B. is an employee of IMMAGINA BioTechnology S.r.l. Alessandro Quattrone: A.Q. is a shareholder of IMMAGINA BioTechnology S.r.l. Graziano Guella: G.G. is shareholders of IMMAGINA BioTechnology S.r.l. Gabriella Viero: G.V. is a scientific advisor of IMMAGINA BioTechnology S.r.l. Massimiliano Clamer: M.C. is the founder of, director of, and a shareholder in IMMAGINA BioTechnology S.r.l., a company engaged in the development of new technologies for gene expression analysis at the ribosomal level. The other authors declare that no competing interests exist.

## Funding

| Funder | Grant reference number | Author |
| --- | --- | --- |
| Autonomous Province of Trento | LP6/99 | Luca Minati<br>Claudia Firrito<br>Alessia Del Piano<br>Alberto Peretti<br>Paola Bernabo<br>Massimiliano Clamer |

The funders had no role in study design, data collection and interpretation, or the decision to submit the work for publication.

## Author contributions

Luca Minati, Conceptualization, Formal analysis, Validation, Investigation, Writing - original draft; Claudia Firrito, Alessia Del Piano, Investigation; Alberto Peretti, Investigation, Methodology, Writing - first workflow and methods; Simone Sidoli, Data curation, Formal analysis, Methodology; Daniele Peroni, Data curation, Formal analysis, Investigation, Methodology; Romina Belli, Data curation, Formal analysis, Visualization, Methodology; Francesco Gandolfi, Data curation, Formal analysis; Alessandro Romanel, Data curation, Formal analysis, Investigation, Visualization, Methodology; Paola Bernabo, Data curation, Project administration; Jacopo Zasso, Visualization, Methodology; Alessandro Quattrone, Methodology; Graziano Guella, Writing - review and editing; Fabio Lauria, Formal analysis; Gabriella Viero, Conceptualization, Data curation, Formal analysis, Supervision, Methodology, Writing - review and editing; Massimiliano Clamer, Conceptualization, Resources, Data curation, Formal analysis, Supervision, Funding acquisition, Validation, Investigation, Visualization, Methodology, Writing - original draft, Project administration, Writing - review and editing

## Author ORCIDs

Daniele Peroni (iD) http://orcid.org/0000-0002-0862-266X
Romina Belli (iD) http://orcid.org/0000-0002-5690-2797
Jacopo Zasso (iD) http://orcid.org/0000-0002-3151-6443
Gabriella Viero (iD) https://orcid.org/0000-0002-6755-285X
Massimiliano Clamer (iD) https://orcid.org/0000-0001-8185-059X

## Decision letter and Author response

Decision letter https://doi.org/10.7554/eLife.59303.sa1
Author response https://doi.org/10.7554/eLife.59303.sa2

# Additional files

## Supplementary files

• Transparent reporting form

## Data availability

All data generated or analysed during this study are included in the manuscript and supporting files. Source data files have been provided. All sequencing data are deposited in public archives and made available upon publication.

The following datasets were generated:

| Author(s) | Year | Dataset title | Dataset URL | Database and Identifier |
| --- | --- | --- | --- | --- |
| Minati L, Romanel A, Peretti A, Gandolfi F, Clamer M | 2021 | One-shot analysis of translated mammalian lncRNAs with AHARIBO - Mass-spectrometry proteomics data | https://www.ebi.ac.uk/pride/archive/ | |
| | | ProteomeXchangeConsortium, PXD022679 | | |

| Minati L, Romanel A, Peretti A, Gandolfi F, Clamer M, Firrito C | 2021 | One-shot analysis of translated mammalian lncRNAs with AHARIBO - Ribo-seq processed data | https://www.ncbi.nlm.nih.gov/geo/ | GEO repository, GSE167865 |
| Minati L, Romanel A, Peretti A, Gandolfi F, Clamer M, Firrito C | 2021 | One-shot analysis of translated mammalian lncRNAs with AHARIBO - NGS processed BAM files | https://www.ncbi.nlm.nih.gov/bioproject/ | NCBI BioProject, PRJNA692822 |

The following previously published datasets were used:

| Author(s) | Year | Dataset title | Dataset URL | Database and Identifier |
| --- | --- | --- | --- | --- |
| Ingolia NT, Lareau LF, Weissman JS | 2011 | Ribosome Profiling of Mouse Embryonic Stem Cells Reveals the Complexity and Dynamics of Mammalian Proteomes | https://www.cell.com/fulltext/S0092-8674%2811%2901192-5#secd1715918e1381 | GEO, GSE30839 |
| van Heesch S, Witte F, Schneider-Lunitz V, Schulz JF, Adami E, Faber AB, Kirchner M, Maatz H, Blachut S, Sandmann C-L, Kanda M, Worth CL, Schafer S, Calviello L, Merriott R, Patone G, Hummel O, Wyler E, Obermayer B, Mücke MB, Lindberg EL, Trnka F, Memczak S, Schilling M, Felkin LE, Barton PJR, Quaife NM, Vanezis K, Diecke S, Mukai M, Mah N, Oh S-J, Kurtz A, Schramm C, Schwinge D, Sebode M, Harakalova M, Asselbergs FW, Vink A, de Weger RA, Viswanathan S, Widjaja AA, Gärtner-Rommel A, Milting H, dos Remedios C, Knosalla C, Mertins P, Landthaler M, Vingron M, Linke WA, Seidman JG, Seidman CE, Rajewsky N, Ohler U, Cook SA, Hubner N | 2019 | The Translational Landscape of the Human Heart | https://www.cell.com/cell/fulltext/S0092-8674(19)30508-2?_returnURL=https%3A%2F%2Flinkinghub.elsevier.com%2Fretrieve%2Fpii%2FS0092867419305082%3Fshowall%3Dtrue | Publicly available and interactive web application for exploring the results of this paper, http://shiny.mdc-berlin.de/cardiac-translatome/ |

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
