## [Decision Letter]

**Acceptance summary:**

Using AHA labeling approach, the authors reported a one-shot approach by combining RNA-seq, Ribo-seq, and LC-MS. The authors provide evidence that this approach helps identification of translatable lncRNAs.

**Decision letter after peer review:**

Thank you for submitting your article "One-shot analysis of translated mammalian lncRNAs with AHARIBO" for consideration by *eLife*. Your article has been reviewed by three peer reviewers, and the evaluation has been overseen by a Reviewing Editor and James Manley as the Senior Editor. The following individual involved in review of your submission has agreed to reveal their identity: John L Rinn (Reviewer #2).

The reviewers have discussed the reviews with one another and the Reviewing Editor has drafted this decision to help you prepare a revised submission.

Summary:

Minati et al. reported a method called AHARIBO to detect RNAs with active translation. Using L-azidohomoalanine (AHA) to label nascent polypeptides, the authors purified the translation complex via click chemistry followed by high-throughput analysis including RNA-seq. LC-MS, and Ribo-seq. The authors compared translatome between ESC and differentiated neurons (EN) using AHARIBO and found some long non-coding RNAs (lncRNAs) that could encode peptides.

The overall methodology is straightforward and useful. The authors claim that AHARIBO is able to distinguish ribosome-associated lncRNAs and ribosome-translating lncRNAs. This is indeed an important question in the field since a growing body of evidence suggests that a substantial amount of lncRNAs contain functional open reading frames. Although the title, Abstract, and Introduction primarily focus on this topic, the result section did not serve this goal at all. A revision needs to address the key issues below, as well as temper claims and list the potential limitations and interpretations in a revised Discussion.

Essential revisions:

1) AHARIBO detection of nascent peptide vs. mature protein association with RNA.Nascent peptide capture does seem to offer a new approach to measure translation. The core strategy is not adequately validated, however, and so it is not clear that these techniques are capturing proteins and RNAs through nascent-chain labeling. Further, individual applications of this technique for proteomics or sequencing are not subject to incisive tests that clearly distinguish the proposed nascent chain capture mechanism from alternative explanations. It is in fact unclear if there is a straightforward path to address these central concerns.

a) It is clear that the great majority of AHA label is found in completed free proteins rather than in nascent proteins (Figure 1B). Capture of ribosomal proteins (along with a range of other unspecified proteins) in AHA- controls doesn't exclude the possibility that this (Figure 1D) reflects enrichment of new proteins. In addition to labeling of mature proteins, it seems that only 2- to 3-fold enrichment is achieved in comparison with AHA- samples (Figure 1—figure supplement 1D). This means that a large fraction of captured protein, including captured ribosomes/polysomes - is unlabelled background.

b) Likewise, AHA labeling is often used to measure nascent protein synthesis, much like pSILAC labeling. What is the evidence that the concordance in pSILAC and AHARIBO-nP doesn't simply reflect the labeling of fairly new but completely synthesized protein in AHARIBO-nP lysates?

c) If AHARIBO is capturing nascent peptides, certain strong polarity effects are expected: peptides should be strongly enriched near the N-terminus of proteins relative to the C-terminus. Ribosome footprints should be absent from the first 30 – 40 codons, because these proteins should not expose nascent peptide.

2) Evidence for newly detected lncRNA encoded peptides via epitope tagging. Can the authors express a tagged version (even if ectopically) of some of the newly identified peptides. This would be a complimentary validation to the mass spec performed and provide spatial localization information. For example, the lncRNA TUG1 was recently reported in Cell and Genome Biology to encode a peptide and is highly abundant in both cell types. Cell : DOI:https://doi.org/10.1016/j.cell.2019.05.010) and genome biology: in press, currently: https://www.biorxiv.org/content/10.1101/562066v1. Do the authors find TUG1 translated in these cells (the RNA is abundant in both) that would be another validation of a "validated" new peptide that is larger than 100a.a

[Editors' note: further revisions were suggested prior to acceptance, as described below.]

Thank you for resubmitting your work entitled "One-shot analysis of translated mammalian lncRNAs with AHARIBO" for further consideration by *eLife*. Your revised article has been evaluated by James Manley (Senior Editor) and a Reviewing Editor.

The manuscript has been improved but there are some remaining issues that need to be addressed before acceptance, as outlined below:

1) The validation of TUG1 peptide strengthens the original conclusion. Since TUG1 cannot be identified from AHARIBO, it will be highly desirable if the authors could show any newly identified peptides from lncRNA, i.e., translatable lincRNA that has not been reported before. At least, a thorough discussion about this potential is needed.

2) The puromycin treatment resulted in ~15-20% reduction in AHARIBO signal rather than the >80% seen in ribosome profiling experiments, which implies that a substantial fraction of the AHARIBO signal comes from nonspecific background. Please incorporate text to explicitly address this point – should puromycin treatment always be done to confirm AHARIBO screen hits?

---

## [Author Response]

Essential revisions:1) AHARIBO detection of nascent peptide vs. mature protein association with RNA.Nascent peptide capture does seem to offer a new approach to measure translation. The core strategy is not adequately validated, however, and so it is not clear that these techniques are capturing proteins and RNAs through nascent-chain labeling. Further, individual applications of this technique for proteomics or sequencing are not subject to incisive tests that clearly distinguish the proposed nascent chain capture mechanism from alternative explanations. It is in fact unclear if there is a straightforward path to address these central concerns.a) It is clear that the great majority of AHA label is found in completed free proteins rather than in nascent proteins (Figure 1B).

We agree with the reviewer that, to some extent, AHA labeled proteins, fully synthesized and released from ribosomes during the incubation time with AHA, could be captured by AHARIBO. This is also mentioned in our original manuscript: “The high signal observed in light fractions is caused by AHA-labeled proteins released from ribosomes”. Proteins fully synthesized could affect AHARIBO’s efficiency in different ways. Therefore, we approached the problem as follow:

1) We check if a sucrose cushioning of the cell lysate could increase AHARIBO’s efficiency in terms of total RNA recovery. Hence, the separation of free full-synthesized proteins from ribosomes before performing AHARIBO, could help to reduce the background.

2) We performed additional controls to further demonstrate the robustness of the method, even without the need of a sucrose cushioning of the cell lysate

1) Sucrose cushioning of the AHARIBO cell lysate

Starting from a HeLa cell lysate, we compared the efficiency of AHARIBO before and after sucrose cushioning in terms of total RNA capture (calculated as the fold change enrichment between AHA treated and control, AHA+/AHA-). As a matter of fact, a clarification of the cell lysate from full length proteins allows better AHARIBO-rC efficiency expressed as RNA recovery.

The cushioning step could help to improve the AHARIBO efficiency in highly proliferative cells, when protein synthesis rate is maximized and a larger amount of protein can be produced and released from ribosomes during AHA incubation. We included this information in the manuscript and in Figure 1—figure supplement 1. It is important to note that the cushioning step can be relevant for the identification of translated lncRNAs. In fact, the differences between AHARIBO and global translatome for the enrichment in lncRNA are more pronounced after sucrose cushioning (Figure 4A) than without sucrose cushioning (Figure 4—figure supplement 1C).

That being said, since (i) AHARIBO is meant to be a fast protocol with low instrumental needs (e.g. ultracentrifuge) and (ii) we further confirmed that a clarification of the cell lysate is not essential in low proliferative cell lines (primary neurons, Author response image 1); for the sake of clarity, we preferred to not include this step in the standard experimental setup of Figure 1.

We included this sentence to the manuscript: “To overcome this problem it is possible to perform a pre-cleaning of the cell lysate by sucrose cushioning. This step can increase the efficiency of total RNA isolation compared with the control (no AHA) (Figure 1—figure supplement 1D).”

2) Additional controls to prove the robustness of the method (sections A, B and C)

A) We reasoned that AHARIBO’s effectiveness could be affected by free AHA-tagged in the cell lysate, if the AHA-tagged proteins are RNA binding proteins (RBPs). In this case, a possibility of co-isolation of RNAs bound to AHA-tagged RBPs cannot be excluded. If this is the case, we should observe a bias toward specific subtypes of isolated mRNAs. In particular, we should expect that RBPs binding sites are significantly overrepresented in the isolated AHARIBO mRNA when compared with the global translatome. To rule out this hypothesis, we analyzed differentially expressed genes during cell differentiation (mESC/EN), searching for specific enrichment of sequences known to be recognized by RNA binding proteins. We screened 1000 differentially expressed mRNA (P value < 0.005) from AHARIBO-rC and from the global translatome, and we mapped the RBP motives in the untranslated regions (UTRs) of the selected transcript (http://aura.science.unitn.it/about/) by using the Atlas of UTR Regulatory Activity (AURA, mouse database) (Dassi et al., 2014).

**Author response table 1. resptable1:** AHARIBO-rC. UTR regulatory elements enrichment

**Regulatory factor/element**	**Regulated query genes**	**Enrichment p-value**	**BH-corrected p-value**
Tardbp	119 ( 11.90 % )	< 1.0E-07	< 1.0E-07
Nova2	175 ( 17.50 % )	< 1.0E-07	< 1.0E-07
Srsf3	201 ( 20.10 % )	< 1.0E-07	< 1.0E-07
Ezh2	159 ( 15.90 % )	1.1E-07	0.00000034
Ptbp2	22 ( 2.20 % )	1.9E-07	0.00000057
Apobec1	21 ( 2.10 % )	6.3E-07	0.0000019
Mbnl2	24 ( 2.40 % )	1.26E-05	0.00003775
Ago	21 ( 2.10 % )	7.72E-05	0.00023166
Srsf2	68 ( 6.80 % )	0.001378	0.00275579
Elavl4	2 ( 0.20 % )	0.001655	0.00331066
Elavl2	2 ( 0.20 % )	0.003259	0.00651716
Elavl1	3 ( 0.30 % )	0.004874	0.00974701
Rbms3	1 ( 0.10 % )	0.023689	0.02368882
Rbm3	15 ( 1.50 % )	0.036423	0.03642291
Rbm8a	1 ( 0.10 % )	0.046817	0.04681703

**Author response table 2. resptable2:** Global translatome. UTR regulatory elements enrichment

**Regulatory factor/element**	**Regulated query genes**	**Enrichment p-value**	**BH-corrected p-value**
Tardbp	96 ( 9.60 % )	< 1.0E-07	< 1.0E-07
Nova2	139 ( 13.90 % )	< 1.0E-07	< 1.0E-07
Srsf3	168 ( 16.80 % )	< 1.0E-07	< 1.0E-07
Srsf2	56 ( 5.60 % )	2.95E-06	1.77E-05
Apobec1	19 ( 1.90 % )	9.13E-06	3.65E-05
Ptbp2	19 ( 1.90 % )	1.08E-05	4.33E-05
Srsf1	59 ( 5.90 % )	7.26E-05	0.0002177
Elavl2	2 ( 0.20 % )	0.00325858	0.00651716
Elavl1	3 ( 0.30 % )	0.0048735	0.00974701
Ago	15 ( 1.50 % )	0.01949888	0.03899777
Rbms3	1 ( 0.10 % )	0.02368882	0.02368882
Rbm8a	1 ( 0.10 % )	0.04681703	0.04681703

We noticed that AHARIBO and global translatome retrieve similar results for highly represented regulatory elements. Therefore, we can exclude any major bias due to the co-isolation of RNAs through AHA-tagged RBPs. The only significant and robust difference between the two datasets is related to the regulator factor *histone-lysine N-methyltransferase, Ezh2*. This protein is present in actively dividing cells, widely expressed in early embryos and in central and peripheral nervous system. Since we did not identified this protein in AHARIBO-nP, we can claim that the protein is not AHA-labelled and the related RNA motive is enriched because it probably binds UTRs of differentially expressed transcripts during translation.

In conclusion, our analysis on UTR regulatory elements did not show a strong difference between the two datasets (AHARIBO and global translatome), suggesting that full length AHA-tagged RNA binding proteins do not affect AHARIBO RNA results.

B) A possibility to reduce the free protein content is to reduce the AHA incubation time. Unfortunately, a compromise in the time of incubation is required when cell lines have different protein synthesis rates. We tested AHARIBO on different cellular models (HeLa, mESC) and differentiation state (mESC differentiate into early neurons). Early neurons (EN) have a protein turnover slower than embryonic stem cells (Author response image 1) and proliferative (HeLa) cells. In this work, we empirically defined the optimal time point for RNA enrichment on AHARIBO beads. To be consistent, we set a constant incubation time of 30 min among all experiments. This time is required to maximizing the recovery of both RNA and ribosomal proteins in HeLa, mESCs, and mESC differentiated in early neurons (Author response image 1). This time point is probably more than enough for a proper nascent protein labeling of human immortalized cancer cells (HeLa) and mESCs (where at 30 min a lot of proteins are produced and released) but it is the minimum requirement for a good efficiency in EN without additional step of sucrose cushioning.

**Author response image 1. sa2fig1:** Total protein staining. Left, total proteins from mESC and early neurons (EN) cell lysates (20 µg total protein for each sample measured by Bradford assay) loaded on a SDS-PAGE and the membrane stained by biotin-cycloaddition followed by streptavidin-HRP. Central and right panel, agarose gel electrophoresis of total RNA extracted from input lysates (1/10 of the total lysate volume) and lysates subjected to AHARIBO pulldown from mESC (left) or EN cells (right) either treated (+) or not (-) treated with AHA.

In conclusion, we provide additional evidences that the enrichment of RNA with AHARIBO is consistent across all biological samples tested.

C) To better characterize AHARIBO for its ability to capture ribosome complexes, we performed the following additional experiments:

i) We inhibit translation with heat shock (10 min at 42°C) during the 30 min of AHA incubation time;

ii) We forced the release of the nascent protein chain from ribosomes by puromycin treatment instead of sBlock.

In both experimental setups we should observe a reduced AHARIBO’s efficiency. In the first case, due to a reduced protein synthesis. In the second case, due to the inhibition of protein synthesis and a release of the nascent chain from the ribosome.

i) Heat shock. First, we checked if the heat shock was effective by performing a RT-qPCR to monitor the RNA level of a member of the 70 kilodalton heat shock proteins (HSPA4 ), a family of proteins known to be involved in the response to heat shock (Richter et al., 2010). We observed that HSPA4 is upregulated in AHARIBO-rC (Author response image 2) for both HeLa and mESC. Then, we checked if the heat shock affects global protein synthesis by SDS-PAGE (Author response image 2). As a consequence of the heat shock, we observed a decrease of the AHA-labelled proteins (representing de-novo protein synthesis) compared with untreated cells (no heat shock).

**Author response image 2. sa2fig2:** a) HSPA4 fold change (ΔΔct) measured by qPCR in mouse embryonic stem cells and HeLa with or without heat shock (10 min at 42°C). For each sample, HSPA5 AHA+/AHA- signal ratios were normalized to the control and to the housekeeping gene. b) Left, quantification of AHA content before (control) and after heat shock. On the right, representative image of a SDS-PAGE reporting the total protein content for heat shock (with and without AHA) and not treated sample (with and without AHA). In each lane a total of 1 ug of protein as measured by Bradford assay was loaded. Staining of the membrane was performed by biotin cycloaddition followed by streptavidin-HRP. Experiments were performed in triplicates.

As reported in the original manuscripts, when protein synthesis is inhibited (e.g. arsenite treatment), AHARIBO-rC captures less ribosomes and associated RNAs. According to that, and after having established that the heat shock condition was effective, we applied AHARIBO-rC on HeLa cells with or without heat shock followed by RNA extraction form AHARIBO-rC. We observed that the total AHARIBO-rC RNA is reduced after heat shock, as expected due to the reduction in protein synthesis.

In conclusion, a down-regulated protein synthesis by heat shock results in a reduction of RNA captured relative to the control, as reported in the main text for the arsenite treatment. We included these results in Figure 1—figure supplement 3.

ii) Puromycin treatment. We performed AHARIBO experiment in HeLa cells treated with 50 μm of puromycin. Puromycin reacts with the nascent chain, terminating translation and causing ribosome disassembling. Therefore, AHA-tagged nascent chains will no longer be available on ribosomes for AHARIBO capture. Strikingly, we observed a strong improvement of the signal-to-background ratio for AHARIBO RNA enrichment as further validated by LC-MS and western blot analysis (see below, answer to reviewer #1b).

In conclusion, the release of the nascent peptide chain from the ribosomes by means of puromycin results in a reduction of RNA captured relative to the control. We included this additional validation of AHARIBO in Figure 1—figure supplement 3.

Capture of ribosomal proteins (along with a range of other unspecified proteins) in AHA- controls doesn't exclude the possibility that this (Figure 1D) reflects enrichment of new proteins. In addition to labeling of mature proteins, it seems that only 2- to 3-fold enrichment is achieved in comparison with AHA- samples (Figure 1—figure supplement 1D). This means that a large fraction of captured protein - including captured ribosomes/polysomes - is unlabelled background.

We thank the reviewer for drawing our attention to the unlabeled background present in AHARIBO. A main source of this background could be generated by unwanted reactions of the biotin-dibenzocyclooctyne (DBCO) reagent in the cell lysate when AHA is not present. In fact, the kinetic of the reaction of DBCO with AHA is fast and efficient, but when AHA is not present (as in the “AHA-” control) the ligand can react more efficiently with other chemical groups (van Geel et al., 2012). To better understand the problem, we performed AHARIBO with two different controls:

*control 1*: AHARIBO with AHA but without DBCO

*control 2*: AHARIBO with AHA but with puromycin treatment instead than sBlock treatment.

As suggested by the reviewer, we strengthened and validated our manuscript by adding new set of experiments (data included in Figure 1—source data 1):

i) New LC-MS analysis with *control 1*

After isolation of the ribosomal complexes with AHARIBO-rC from HeLa cell lysates, we extracted all proteins captured. In parallel the same experiment was performed without biotin-DBCO, therefore no specific pull-down of AHA-tagged nascent peptides and ribosomes is possible. After trypsin digestion and LC-MS analysis on all proteins co-purified with AHARIBO-rC we selectively identified all ribosomal proteins. We observed from a 4-fold (e.g. RPL23A) to 18-fold (e.g. RPL4, RPL7) enrichment is achieved for AHARIBO-rC in comparison with “control 1” samples. This means that a portion of the unlabeled background is DBCO related.

ii) New LC-MS analysis and western blots results with *control 2*

We reasoned that an additional experiment to account for the DBCO-related background is to include a control where the nascent peptide is released from ribosome instead of being stabilized on it with sBlock. We treated cells with puromycin instead of sBlock and we repeated the LC-MS analysis (in this case AHA is present in both control and puromycin treated sample). Our results show that less background and a stronger enrichment is obtained in LC-MS and in western blot data compared with the AHA^-^ control (no AHA). This result is in agreement with our data on the RNA recovery reported at the reviewer answer 2cii.

iii) Dedicated experiments with an ectopically expressed micropeptide.

See answer to reviewer #2. The ectopic expression of the short peptide Tug1-Boat demonstrates that in the absence of methionine in the nascent peptide chain, the Tug1-Boat transcript is not efficiently co-purified with AHARIBO (50-fold signal-to-background).

Although the DBCO-related background is hardly removable, these additional experiments confirm the robustness of the method. We changed Figure 1 and the main text according to the new results and we added this to Figure 1 and Figure1—figure supplement 2.

b) Likewise, AHA labeling is often used to measure nascent protein synthesis, much like pSILAC labeling. What is the evidence that the concordance in pSILAC and AHARIBO-nP doesn't simply reflect the labeling of fairly new but completely synthesized protein in AHARIBO-nP lysates?

The reviewer is correct in highlighting the relation between pSILAC and AHARIBO-nP. We apologize that this point was not clear enough.

We tackled the question comparing 100 differentially expressed proteins (mean of triplicate sets of data) detected by pSILAC and AHARIBO-nP, whose transcripts were detected by AHARIBOrC RNA as well. Our data show a poor concordance between the two datasets (Author response image 3). Therefore, AHARIBO-nP is not reflecting the total protein turnover as measured by pSILAC, but it is capturing only a fraction of the pSILAC proteome.

**Author response image 3. sa2fig3:** Protein fold change (log2) of differentially expressed proteins during cellular differentiation of mESCs to early neurons. Comparison between pSILAC(black) and AHARIBO proteome (red).

Additionally, we compared differentially expressed genes detected using the global translatome versus the protein turnover (pSILAC). The analysis revealed a poor concordance between pSILAC and global translatome RNA variations (Pearson’s r = 0.27), differently from what we observed with AHARIBO (main Figure 3).

Therefore, we can conclude that AHARIBO protein does not mostly reflect all pSILAC data, but only a fraction of the them represented by translated mRNA during AHA incubation. We modified the Results section and we add to Figure 3—figure supplement 1.

c) If AHARIBO is capturing nascent peptides, certain strong polarity effects are expected: peptides should be strongly enriched near the N-terminus of proteins relative to the C-terminus. Ribosome footprints should be absent from the first 30 – 40 codons, because these proteins should not expose nascent peptide.

We thank the reviewer for suggesting the possibility of a polarity in our ribosome profiling data. Related to that, other groups reported a reduced coverage of ribosome footprints at the beginning of the transcript in selective ribosome profiling experiments to capture ribosomes engaged with chaperon proteins. For example, the Bakau’s lab (Figure 3A in Oh et al., 2011) reported that the *E. coli* trigger factor (TF) engages with the polypeptide emerging from the exit channel after the first 100 amino acids are translated. A similar result was observed in yeast (Figure 1B in Döring et al., 2017) where a selective Ribo-seq revealed a reduced read density in the first 50 codons reflecting the interaction of chaperon proteins with the nascent chains beyond a length of 50 aa.

Given that the first 30 aa are buried inside the ribosome exit tunnel and chaperon proteins need a certain number of amino acid to support the proper folding of the newly synthetized polypeptide, it is clear that these proteins engage with nascent chain when at least 20 residues are exposed from the exit tunnel of the ribosome.

To understand if this was the case for AHARIBO as well, we analyzed the differences between the P-site density of AHARIBO Ribo-seq and total RIBO-seq (input).

First, we looked at the ratio between the average P-sites density distribution in AHARIBO RIBO-seq and the average number of P-sites on the whole CDS, in the first 120 nucleotides from initiation. Strikingly, we observed a significant (measured with two-sided Wilcoxon– Mann–Whitney test) lower P-site density for AHARIBO in the first 120 nt, confirming that the first 40 amino acids are less covered by reads (Author response image 4, left).

**Author response image 4. sa2fig4:** Left, P-site density initiation/CDS ratio for total RIBO-seq (input) and AHARIBO Ribo-seq. Right, metagene profiles showing the ratio between the AHARIBO density and input read density within the first 450 codons from the start codon. Violet arrow, first 25 codons from the start codon. Blue arrow, first 50 codons from the start codon.

Then, we looked at the ratio between AHARIBO and Input metagene profiles, and we observed a reduction of the RPF signal in the first 50 codons, with the strongest reduction in the first 25 codons (Author response image 4, right).

These results confirm that:

1) In agreement with the hypothesis of an enrichment of the signal near the N-terminus, we observed a reduced RPF density in the first 120 nt in AHARIBO when compared to the input.

2) The reduction of signal is not broadly distributed in the first 100 codons as reported for chaperon proteins, but mainly concentrated in the first 25-50 codons. This could be explained by the different type of engagement with the nascent chain. Unlike experiments published for chaperon proteins, AHARIBO captures the nascent chain through a small molecule, i.e. AHA reaction with biotin-DBCO followed by beads pulldown. We cannot exclude that the small biotin-DBCO molecule reacts with the N-terminal AHA-tagged nascent chain just outside the exit tunnel, or partially inside the main exit tunnel. According to that, we speculate that the molecule could bind the nascent chain, much before large chaperon proteins interact with the nascent chain.

In support of this hypothesis, we observed that Joachim Frank’s lab (Gabashvili et al., 2001) and Song’s lab (Dao Duc et al., 2019) reports that the ribosome exit tunnel have a dynamic behavior and multiple exits to facilitate the extrusion of nascent peptides (Figure 1 in Duc DK et al., 2019) from ribosomes. In particular, by comparing 20 different cryo EM and X-ray crystallography structure of the ribosome from all three domains of life, D. Duc. et al. observed that the first 25-30 aa buried inside the ribosome exit tunnel are passing through two main constrictions. A first, at about 30 Å (about 9 aa) from the PTC and made by the uL4 and uL22 protein loops. The second, at about 50 Å (about 15 aa) from the PTC formed by an extended arm of uL4 (Figure 4A in Duc DK et al., 2019). The radius of the constriction is about 4 Å at the first troughs and 3-4 Å at the second (in human). While the radius at the exit is about 7-8 Å.

With a length of 32 Å and a radius of 7 Å, biotin-DBCO can potentially reach the internal part of the tunnel at about 60-80 Å from the PTC. This ribosome architecture at the exit tunnel may influence the interaction of small molecules with the ribosomes. Therefore, we should expect a reduction of the Ribo-seq signal in the first 75 nt from the start codon. This is in agreement with our meta-profile reported in Author response image 4 (right) and with the Tug1-boat micropeptide ectopic expression experiment (Reviewer answer #2).

3) We still observed some signal at the start codon, although less than in the input. This can be related to the background or to a possible capture of the tRNAi^met^ (substituted by tRNAi^AHA^ in AHARIBO) within the 48S translational initiation complex. This complex could generate signal at the start codon as previously reported for selective 48S profiling experiments (Bohlen et al., 2020).

At the present stage, the hypothesis on ribosome profiling data reported above are not supported by additional biochemical and structural data. For the sake of clarity we did not include these information in the manuscript. Nevertheless, we further discussed this point in the answer to reviewers essential revision 2.

We are aware that AHARIBO for ribosome profiling could have a better mechanistic description, but this topic is out of the scope of this work and could result in a separate publication.

2) Evidence for newly detected lncRNA encoded peptides via epitope tagging. Can the authors express a tagged version (even if ectopically) of some of the newly identified peptides. This would be a complimentary validation to the mass spec performed and provide spatial localization information. For example, the lncRNA TUG1 was recently reported in Cell and Genome Biology to encode a peptide and is highly abundant in both cell types. Cell : DOI:https://doi.org/10.1016/j.cell.2019.05.010) and genome biology: in press, currently: https://www.biorxiv.org/content/10.1101/562066v1. Do the authors find TUG1 translated in these cells (the RNA is abundant in both) that would be another validation of a "validated" new peptide that is larger than 100a.a

The suggestion of the reviewer is extremely useful for a better understanding of AHARIBO mechanism and efficiency. Unfortunately, we did not find TUG1 transcripts in our mESC and EN RNA-seq and Ribo-seq datasets. Hence, a proper discussion about the capture TUG1 with AHARIBO in mESC and EN is not possible.

We, therefore, decided to ectopically express the putative protein produced by the open reading frame of TUG1, called TUG1-BOAT (Tug1-Bifunctional ORF and Transcript) in HeLa cells as reported in (Lewandowski et al., 2020).

Construct generation and transfection was performed as in Lewandowski J.P. et al., 2019 with some minor changes (reported in the updated Materials and methods section of the manuscript) to adapt the experimental setup to the AHARIBO method. Briefly, we synthesized three different constructs for human Tug1 ORF1 (Thermo Scientific):

1) The first (called *WT TUG1-BOAT*) is the one reported in Lewandowski J.P. et al., 2019. It has a non-canonical start codon and a methionine at 75 nt (25 aa) upstream the stop codon.

2) The second (called ΔTUG1-BOAT), is deleted by the only methionine of the sequence present at 75 nt upstream the stop codon

3) The third (called +1Met TUG1-BOAT), has the non-canonical CTG start codon exchange with an ATG codon (methionine). Therefore, the third construct has two methionines, one at the N terminal and a second one at 25 aa before (75 nt) the stop.

We performed target TUG1-BOAT qPCR analysis on AHARIBO pull-down and input for each experiment/construct after 12 and 24 hours of transfection, to validate the efficiency in capturing short translated ORF deriving from a lncRNA (TUG1).

Our RT-qPCR analysis shows a good efficiency (10x – 50x compared with the control) of AHARIBO in capturing TUG1-BOAT RNA when a methionine at the N terminal of the encoded peptide is present (about 50 times more in +1Met TUG1-BOAT than with ΔTUG1-BOAT).

We observed that the WT transcript is captured as well, although the methionine is only 25 aa far from the stop codon. The 25 amino acids are just enough for the polypeptide to protrude from the exit tunnel (25 aa = 75 nt = about 90 Å) This is in agreement with our ribosome profiling data (showing a lower density in the first 25 codons) confirming that AHARIBO can engage with short nascent peptides at the exit of the ribosome tunnel (answer to reviewers essential revision #1c).

The expression of the exogenous small ORF TUG1-BOAT is a valuable demonstration of the AHARIBO efficiency. We described this result in the main text of the manuscript and we included this in Figure 1—figure supplement 3.

Next, to gain more insights into the function of thee lncRNAs in mESCs and early neurons we:

i) Compared our results with a dataset of experimentally validated lncRNA producing short peptides (van Heesch et al., 2019) in rat and mouse heart.

We noted that 19 lncRNAs detected by AHARIBO-rC in mESCs, are present in the dataset reported by Heesch S. et al., 2019, highlighting the fact that AHARIBO can capture translated lncRNA. Since most of these lncRNA are tissue specific (Heesch S et al., 2019), we expected that only a small subset of mESC and EN lncRNAs captured by AHARIBO were in common with the lncRNAs identified by Heesch S et al. in rat and mouse heart.

ii) Analyzed the lncRNA differentially expression level after heat shock in mESCs

We exposed mESCs to a short heat shock (42°C for 10°C), to induce the expression of a subset of genes involved in the adaptive response. First, we confirmed that cells react to heat shock by monitoring the upregulation of the heat shock transcript (HSPA4). Next, we confirmed that AHARIBO can successfully detect changes induced by the stress (see answer reviewer #1).

Then, we performed AHARIBO-rC RNA-seq in mESCs before and after heat shock and we analyzed differentially expressed lncRNAs. We did not observe any significant change in lncRNAs differential expression at the level of the global translatome, while in AHARIBO-rC we detected 11 lncRNA (Table 3) differentially expressed after heat shock. Unfortunately, none of them has been validated by other groups or is known to produce micropeptides. Therefore, we decided to not include these data in the manuscripts, but they further demonstrate the potential of our technique. In fact, this experiments could help further investigation on the effect of lncRNAs in heat shock, as previously reported on other species (Bernabò et al., 2020), since only a few data related to human cells are published so far (Place and Noonan, 2014).

Additional proteomic analysis are needed to better characterized potentially translated lncRNA after heat shock in human cells, which may result in a separate work.

**Author response table 3. resptable3:** lncRNA differentially expressed in mESC after heat shock and captured by AHARIBO-rC.

geneSymbol	logFC	logCPM	PValue
Gm28592	4.045326	-0.18544	1.01E-07
Gm26635	2.237428	0.055059	7.37E-08
1110019D14Rik	-2.13009	3.224659	4.19E-07
4930467D21Rik	-2.46151	2.567217	2.90E-07
Gm26776	-2.6145	0.752226	1.83E-07
Malat1	-2.65273	11.84434	1.38E-08
AC162181.1	-2.81543	1.301371	1.40E-10
Gm30551	-3.16013	0.589125	4.86E-09
Gm5432	-3.70264	-0.11255	1.06E-07
mt-Rnr2	-4.56884	15.37853	2.27E-10
4930440I19Rik	-4.61188	0.014935	3.90E-07

[Editors' note: further revisions were suggested prior to acceptance, as described below.]

The manuscript has been improved but there are some remaining issues that need to be addressed before acceptance, as outlined below:1) The validation of TUG1 peptide strengthens the original conclusion. Since TUG1 cannot be identified from AHARIBO, it will be highly desirable if the authors could show any newly identified peptides from lncRNA, i.e., translatable lincRNA that has not been reported before. At least, a thorough discussion about this potential is needed.

We thank the reviewer for having appreciated our additional experiments and for the constructive comment and thorough suggestion about the TUG1 results.

We addressed the question raised by the reviewer through additional analysis of our AHARIBO RNA-seq, RIBO-seq and proteomic data.

In AHARIBO RIBO-seq, we identified numerous lncRNAs covered by ribosome footprints. By intersecting AHARIBO RIBO-seq data with those obtained from standard methods (RIBO-seq and RNA-seq after sucrose cushioning) and AHARIBO-rC RNA-seq, we identified more than hundred common putative translated lncRNAs (Figure 4B, new version of the manuscript). To validate the coding potential of lncRNAs identified by AHARIBO RNA-seq (Figure 4D, new version of the manuscript) and Ribo-seq, we translated *in silico* the transcripts in all frames to find potential ORFs. *In silico* peptides were used to match confident matching spectra obtained from the AHARIBO-nP protein dataset. We found 129 lncRNAs in common between all AHARIBO protocols and standard RIBO-seq (Figure 4D). Among these lncRNAs we identified a micropeptide (Mm47) of 47 aa (Figure 4D, new version of the manuscript) with a high degree of confidence in mass spectroscopy data. According with our hypothesis and results, this micropeptide was recently discovered and characterized by Bhatta and colleagues (Bhatta et al., 2020).

Several other peptides show high confidence of translation events with *in silico* prediction. Even if they were not perfectly matching our proteomic spectra, these peptides are likely generated from lncRNA not known to be translated. These list of transcripts is reported in Figure 4—source data 3 and listed below (in brackets the unique putative peptide identified).

List of matching peptides from lncRNAs as “high” confidence matching spectra:

ENSMUST00000051089 (NSFVNDIFER), ENSMUST00000181328 (KIDNQINLPK), ENSMUST00000181149 (KINQLQNMVKDNK), ENSMUST00000099446 (NLMNVINVVKLLHFS), ENSMUST00000180524 (MSPSQLLELKRNQ), ENSMUST00000182499 (VCVALIINICHIMI), ENSMUST00000134140 (NGGGLLMSYVIK), ENSMUST00000180432 (ELAEQPSSALKTSNREQ), ENSMUST00000181251 (QLTDNQRVNQKA), ENSMUST00000179344 (KELQLK), ENSMUST00000181443 (KGPNDISLAQSYLPI), ENSMUST00000071101 (KNNPPPQNAKPK), ENSMUST00000180407 (IELRENLQTY), ENSMUST00000180489 (EISASANLELNGAPSQQ), ENSMUST00000188038 (LALEELR), ENSMUST00000149246 (LLLPGVIK).

In addition, six abundant lncRNAs identified in AHARIBO RNA-seq show massive ribosome occupancy in our AHARIBO RIBO-seq data ( > 99th percentile of all lncRNA identified by AHARIBO RIBO-seq). *In silico*-translation of these peptides results in many potential micropeptids (ENSMUST00000180396 (23), ENSMUST00000181751 (61), ENSMUST00000182010 (43), ENSMUST00000192833 (94), ENSMUST00000200021 (27), ENSMUST00000223012 (86)

To further strengthen our results, we analysed the ribosome occupancy profile of three lncRNAs, which display hallmarks of translation in AHARIBO RIBO-seq and the predicted micropeptides position along the transcript in the three different frames of transation. None of them are annotated as translated lncRNA.

As requested by the reviewer, we add the results shown in a new supplementary figure (Figure 4—figure supplement 3) and modify the manuscript text as follow in the Results paragraph:

“Several lncRNA show high confidence of translation events with in silico prediction, even if they were not perfectly matching our proteomic spectra (Figure 4—figure supplement 3), paving the way for a better characterization of translatable lncRNA that has not been reported before.”

and in the Discussion:

“To overcome existing limitation in LC-MS detection, many other translation events on lncRNAs can be predicted combining AHARIBO approaches with in-silico translation of the identified leads. This approach would likely allow to selectively validate a list of still uncharacterized lncRNAs.”

2) The puromycin treatment resulted in ~15-20% reduction in AHARIBO signal rather than the >80% seen in ribosome profiling experiments, which implies that a substantial fraction of the AHARIBO signal comes from nonspecific background. Please incorporate text to explicitly address this point – should puromycin treatment always be done to confirm AHARIBO screen hits?

We thank the reviewer for his/her suggestion. To better understand how much signal is caused by nonspecific background, we performed a set of experiments with puromycin treatment before applying AHARIBO. In particular:

1) we compared AHARIBO’s efficiency in isolating total RNA with or without puromycin pre-treatment. We observed a 12-fold signal-to-background in puromycin treat sample relative to control (Figure 1—figure supplement 3).

2) we compared AHARIBO’s efficiency in isolating proteins with or without puromycin pre-treatment. By both LC-MS analysis and immunoblotting we identified ribosomal proteins among of all proteins captured by AHARIBO. Our proteomic results show a 1.2- to 2-fold signal-to-background in puromycin treated sample compared with the control; i.e not puromycin treated (AHARIBO-rC/AHA+PURO). Immunoblotting performed to analyse two selected ribosomal protein shows a 2- to 9-fold signal-to-background in puromycin treat sample relative to control (Figure 1—figure supplement 2).

Overall, RNA detection methods (RNA-seq, RIBO-seq, Syber gold gel staining of total RNA) were more sensitive than LC-MS detection. This may explain the better results seen in ribosome profiling experiments mentioned by the reviewer.

We agree that a fraction of the AHARIBO signal is not specific background. By performing a sucrose cushioning step before applying AHARIBO, it is possible to reduce the background caused by peptides not associated to ribosomes, as shown in Figure 1—figure supplement 1D. We discuss about this possible strategy in the Results paragraph as follows:

“To overcome this problem it is possible to perform a pre-cleaning of the cell lysate by sucrose cushioning. This step can increase the efficiency of total RNA isolation compared with the control (no AHA) (Figure 1—figure supplement 1D)”.

We agree with the reviewer that a puromycin control could be useful control to account for the nonspecific background in proteomic analysis. Therefore, we modified the Discussion of the manuscript accordingly to the reviewer suggestion:

Although the unlabelled background cannot be avoided, a pre-cleaning of the cell lysate with a cushioning step can help to increase the resolution with difficult samples.

“Moreover, a puromycin treatment instead of sBlock could be added as control in proteomic experiments.”.